# Sinus venosus adaptation models prolonged cardiovascular disease and reveals insights into evolutionary transitions of the vertebrate heart

Jacob T. Gafranek[1,2], Enrico D'Aniello[3], Padmapriyadarshini Ravisankar[2], Kairavee Thakkar [4,5], Ronald J. Vagnozzi [6], Hee-Woong Lim [4,7], Nathan Salomonis [4,7] & Joshua S. Waxman [2,7,8] ✉

How two-chambered hearts in basal vertebrates have evolved from single-chamber hearts found in ancestral chordates remains unclear. Here, we show that the teleost sinus venosus (SV) is a chamber-like vessel comprised of an outer layer of smooth muscle cells. We find that in adult zebrafish *nr2f1a* mutants, which lack atria, the SV comes to physically resemble the thicker bulbus arteriosus (BA) at the arterial pole of the heart through an adaptive, hypertensive response involving smooth muscle proliferation due to aberrant hemodynamic flow. Single cell transcriptomics show that smooth muscle and endothelial cell populations within the adapting SV also take on arterial signatures. Bulk transcriptomics of the blood sinuses flanking the tunicate heart reinforce a model of greater equivalency in ancestral chordate BA and SV precursors. Our data simultaneously reveal that secondary complications from congenital heart defects can develop in adult zebrafish similar to those in humans and that the foundation of equivalency between flanking auxiliary vessels may remain latent within basal vertebrate hearts.

The initial evolutionary transitions that resulted in a unidirectional multichambered heart accompanied by specialized arterial and venous populations in vertebrates remain largely unclear. The most parsimonious model for cardiac chamber expansion situates a common chordate ancestor exhibiting a urochordate-like heart with a single distinct myocardial chamber, exemplified by the popular laboratory model *Ciona robusta*, as the branching point from which vertebrates arose[1–3]. Subsequently, the addition of cardiac chambers within

vertebrates parallels the selective pressure that arose with air-breathing organisms having to manage a dual circulatory system and increased oxygen demands[4–6]. Hence, basal (early-branching) vertebrates, such as agnathans (lamprey and hagfish), as well as all fish possess hearts with two cardiac chambers, amphibians and some reptiles have three (two atria and a single ventricle), while birds and mammals have four cardiac chambers (two atria and two ventricles)[7]. Despite having established a scaffold for explaining the general

[1]Molecular and Developmental Biology Graduate Program, University of Cincinnati College of Medicine, Cincinnati, OH 45267, USA. [2]Division of Molecular Cardiovascular Biology and Heart Institute, Cincinnati Children's Hospital Medical Center, Cincinnati, OH 45229, USA. [3]Department of Biology and Evolution of Marine Organisms, Stazione Zoologica Anton Dohrn, 80121 Napoli, Italy. [4]Division of Biomedical Informatics, Cincinnati Children's Hospital Medical Center, Cincinnati, OH 45229, USA. [5]Department of Pharmacology and Systems Physiology, University of Cincinnati, College of Medicine, Cincinnati, OH 45267, USA. [6]Division of Cardiology, Gates Center for Regenerative Medicine, Consortium for Fibrosis Research and Translation (CFReT), University of Colorado Anschutz Medical Campus, Aurora, CO 80045, USA. [7]Department of Pediatrics, University of Cincinnati, College of Medicine, Cincinnati, OH 45267, USA. [8]Division of Developmental Biology, Cincinnati Children's Hospital Medical Center, Cincinnati, OH 45229, USA. ✉e-mail: joshua.waxman@cchmc.org

process of contractile subunit addition in later vertebrate evolution, the auxiliary chambers at the arterial and venous poles of the heart, the bulbus arteriosus (BA) and sinus venosus (SV) in fish, that respectively facilitate blood flow out of and into the contractile cardiac chambers appear to show significant morphological variability within vertebrates[8,9]. Therefore, the dynamics and origins of auxiliary tissues within vertebrate hearts remain unclear.

Regardless of the number of chambers within an organism, normal heart development requires the coordinated integration of a multitude of transcriptional effectors and signaling pathways to establish appropriate cardiac cell populations[10–13]. Furthermore, this intricacy of heart development is evinced in human pathology, as congenital heart defects (CHDs) are the most common congenital malformations globally[14–16]. The prevalence of patients harboring CHDs conveys a significant clinical burden due to limited treatment options apart from surgical intervention[17,18]. Fundamentally, one of the most important processes during heart development is the proper apportionment of atrial and ventricular cardiomyocytes (CMs) from the progressively differentiating first and second heart field (SHF) populations[19]. Aberrant CM identity determinization can be a significant contributor to structural CHDs that affect both the ventricle and atrium[20]. Importantly, secondary complications, such as hypertension-induced vascular remodeling, cardiomyopathy, and heart failure, can occur throughout life in patients with CHDs that have gone undiagnosed, as well as those that have been surgically repaired[21,22]. More than 20% of reported CHD cases specifically afflict the atrium, including atrial septal defects (ASDs)[16,23], and although ventricular malformation prevalence remains static as patients age, the rate of atrial defects observed in adults with CHDs increases[24]. Therefore, adults with CHDs affecting the atrium are disproportionately untreated until later in life because they appear to be able to cope with these malformations. However, these structural defects may result in prolonged cardiovascular stress that emerges as pleiotropic secondary complications leading to morbidity and death[25].

Nr2fs (formerly Coup-tfs), highly conserved nuclear hormone receptor transcription factors, are master regulators of atrial chamber identity within vertebrates[26–28]. Mutations in NR2F2 are associated with patients harboring atrioventricular septal defects (AVSDs)[29–32], classified as a type of ASD[33]. Within the heart, NR2F2 in humans is preferentially expressed in atrial CMs. Work in human embryonic stem cell- and induced pluripotent stem cell-derived CMs supports it is a regulator of atrial CM differentiation[34–36]. Consistent with promoting atrial differentiation in human CMs, *Nr2f2* null mice have shown that Nr2f2 is indispensable for establishing atrial CM identity, as they exhibit a severely diminished common atrial chamber as early as embryonic day (E) 9.5, ultimately causing lethality at E10[37]. Remarkably, in mice when Nr2f2 is conditionally knocked out in CMs, atrial identity is lost, and the cells instead begin to express genes associated with ventricular CM fate by E10.5, with the ventricular gene expression in atrial CM increasing progressively into adulthood[28]. Recently, we reported that zebrafish Nr2f1a is the functional equivalent of mammalian Nr2f2 with respect to heart development in that it is required to promote atrial CM differentiation[38]. Hence, Nr2f TFs are vital, conserved factors that are necessary to convey vertebrate atrial CM identity during embryonic development and maintain identity into adulthood.

Here, using a mutant allele for *nr2f1a* we show that a significant proportion of these fish can survive to adulthood despite their embryonic atrial chamber defects. In adult *nr2f1a* mutants, whose hearts we found effectively lack atria, the SV, the chamber-like venous inflow tract (IFT) of which there is little molecular and cellular understanding, undergoes dramatic morphological remodeling characterized by wall thickening. In-depth cellular and molecular characterization of the zebrafish SV from both wild-type (WT) and *nr2f1a* mutant demonstrates the SV is comprised of an outer smooth

muscle cell (SMC) population and neural crested-derived cells, a composition comparable to the BA at the arterial pole. Moreover, we find the adaptive remodeling observed in the *nr2f1a* mutant SV is due to the proliferation of SMCs, similar to remodeling found with vascular hypertension[39], and can be ameliorated with pharmacological vasodilators or exacerbated by exercise-induced increased blood flow/sheer stress. In addition to the morphological thickening of the SV, our single cell RNA sequencing (scRNA-seq) revealed the adaptive thickening in the *nr2f1a* SV is associated with a genetic shift of some SMC and EC populations toward arterial identity. Transcriptomic profiling of the adult tunicate *Ciona* blood sinuses (BS), which flank a single myocardial chamber, showed symmetric expression of homologous genes found in the SV and BA of the zebrafish heart. Thus, our data demonstrate vascular adaptations from prolonged hemodynamic stress that occur in adult zebrafish with atrial CHDs reflect those found in human patients, as well as may illuminate vital clues for understanding the origins of auxiliary chamber-like vessels in vertebrate heart evolution.

## Results

### The SV is comprised of an outer layer of smooth muscle

Previously, we reported that engineered *nr2f1a* mutant alleles, generated by targeting the coding region, have smaller atria due to reduced differentiation of atrial CMs[38]. In a mutagenesis screen, we identified an *nr2f1a* allele referred to as *acorn worm* (*aco*), henceforth referred to as *nr2f1a^aco^*. This allele harbors a 45 base-pair (bp) deletion (58 bp deletion plus 13 bp insertion) in the *nr2f1a* 5′-untranslated region (UTR), which we predict causes loss of *nr2f1a* translation, as the mutant fish have decreased Nr2f1a expression and indistinguishable embryonic cardiac defects as reported *nr2f1a* mutant alleles (Supplementary Fig. 1a–f)[38]. Surprisingly, unlike the previously reported *nr2f1a* alleles made by targeted genomic editing, we found that the *nr2f1a^aco^* allele exhibits higher viability and can be maintained in homozygosity, suggesting that despite the embryonic atrial chamber defects it may be hypomorphic relative to the engineered alleles. However, we presently cannot rule out other explanations for the greater viability of this allele, such as variations in genetic background. Nevertheless, typically ~50% (53/100 from one representative clutch) of the *nr2f1a^aco^* fish spawned from homozygous adults survive to adulthood and upon reaching maturity initially, typically show no overt signs of distress. Yet, all adult *nr2f1a^aco^* mutants eventually begin to exhibit indicators of cardiac dysfunction, such as pericardial edema and intrathoracic blood pooling (Supplementary Fig. 2a, b).

Dissecting whole hearts from size-matched adult fish revealed that, regardless of whether or not they have yet developed overt pericardial edema, *nr2f1a^aco^* mutants have enlarged ventricles compared to WT controls and exhibit dysmorphic chambers at the venous pole that we initially presumed to be the "atria" (Fig. 1a). However, acid fuchsin orange G (AFOG) staining of sectioned hearts showed that, whereas both cardiac chambers stained for cardiac muscle in the WT control hearts, what we originally had presumed to be "atria" in the *nr2f1a^aco^* mutant hearts actually lacked staining for cardiac muscle (Fig. 1b). Intriguingly, the *nr2f1a^aco^* hearts still seemed to present with a characteristic narrowing at the atrioventricular canal (AVC) between the enlarged ventricle and the presumed "atrial" chamber at the venous pole, which was evident by prominent collagen staining in the valves. However, the enlarged structure directly apposed to the ventricle in *nr2f1a^aco^* mutant hearts was highly collagenous and internally smooth, rather than the expected pectinate myocardium found in WT atria[40,41].

Nonetheless, given the anatomical position of this collagenous structure that we initially presumed was the "atrium," we sought to determine if the tissue expressed the canonical atrial CM sarcomeric marker *atrial myosin heavy chain* (*amhc*; also called *myh6*). In situ hybridization (ISH) and immunohistochemistry (IHC) both revealed

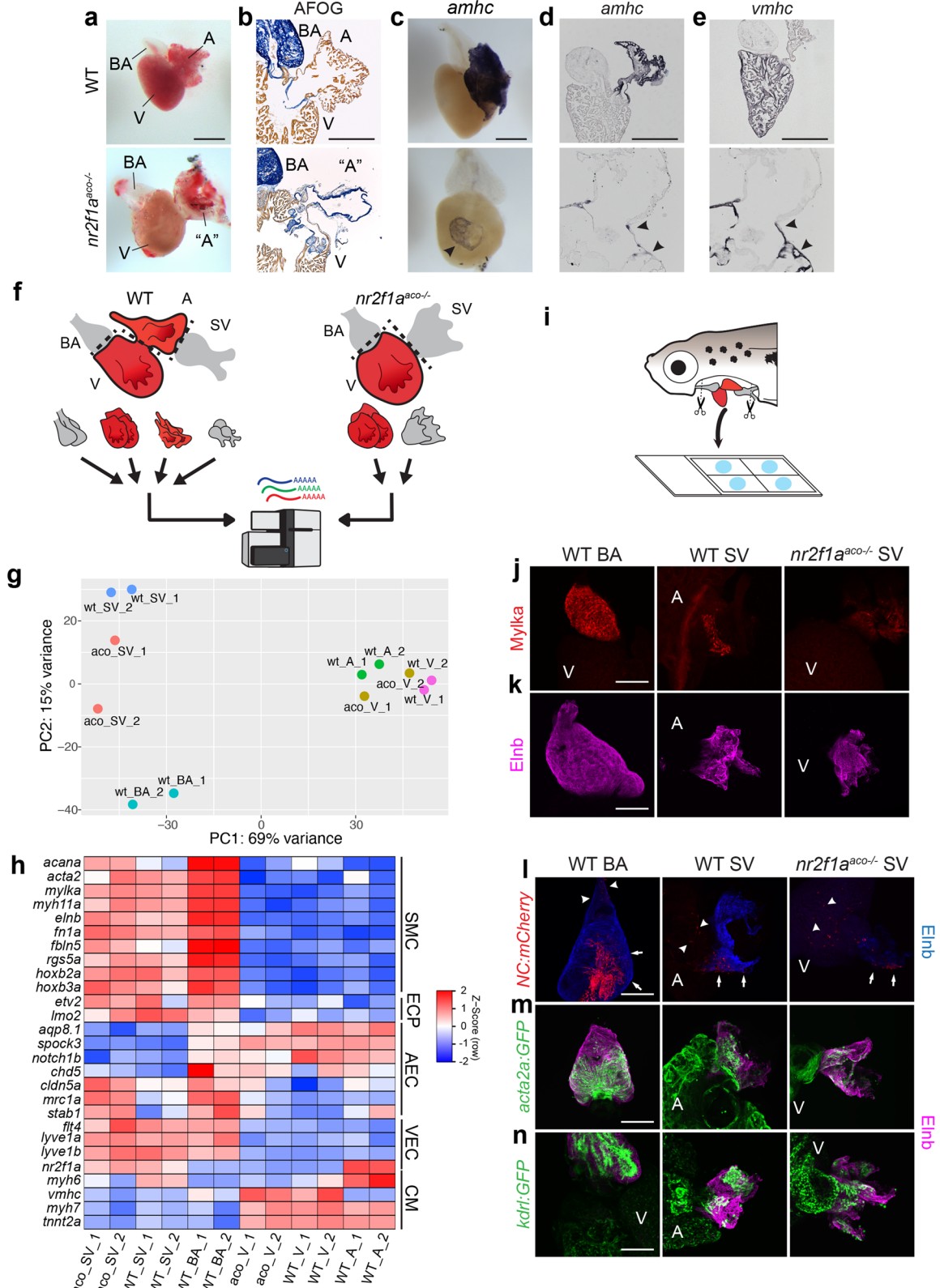

that *nr2f1a^aco* mutant hearts only express the atrial CM marker in a small ring of cells at the venous pole of the ventricle adjacent to the atrioventricular valves (Fig. 1c, d). Notably, these *amhc*⁺ cells also co-expresses the ventricular-marker *ventricular myosin heavy chain* (*vmhc*; also called *myh7*) (Fig. 1e, Supplementary Fig. 3), suggesting these CMs have in part acquired ventricular identity, a phenotypic characteristic highly reminiscent of what has been reported for cardiac

KO of *Nr2f2* in mice[28]. Thus, our data suggest that the adult *nr2f1a^aco* hearts effectively lack atria, yet have large collagenous, chamber-like structures at the venous poles of their hearts.

Given the enlarged, dysmorphic structures that we initially presumed to be the "atria" in the *nr2f1a^aco* mutant hearts were devoid of atrial characteristics, we reasoned that they may instead be part of, or derived from, the venous inflow vasculature, and specifically the SV

**Fig. 1 | Inflow tract tissue adjacent to the venous pole of the heart tissue expands in adult *nr2f1a* mutants lacking an atrium. a** Brightfield images of whole hearts from adult WT and *nr2f1a^aco−/−^* fish. Bulbus arteriosus (BA), Ventricle (V), Atrium (A), putative Atrium ("A"). **b** Brightfield images of sectioned hearts stained with AFOG. Cardiac muscle – brownish-orange, Collagen - blue, fibrin - red. **c** ISH for *amhc* on adult WT and *nr2f1a^aco^* mutant hearts. *Amhc* (arrowhead) in *nr2f1a^aco^* mutant hearts is a relatively small ring adjacent to the ventricle. **d, e** ISH for *amhc* and *vmhc* on sectioned hearts. A restricted ring of CMs expressed both these atrial and ventricular differentiation marker genes adjacent to the venous pole of the heart (black arrowheads) **f**. Schematic depicting subunits of the zebrafish hearts harvested for bulk RNA-seq. **g** PCA analysis from the bulk RNA-seq BA, SV, A, and V samples. The SV from *nr2f1a^aco^* mutant fish occupies space between the WT SV and BA. **h** Heatmap of representative marker genes summarizing SMC, EC progenitor (ECP), arterial EC (AEC), venous EC (VEC), and cardiomyocyte (CM) markers. **i** Schematic of the preparation procedure of juvenile hearts for wholemount IHC. **j, k** Confocal images of wholemount IHC of SMC markers Myosin light chain kinase a (Mylka - red) and Elastin b (Elnb - magenta) in 28 dpf hearts. **l–n** Confocal images of wholemount IHC of BA and SV with transgenic reporters for NC-derived cells *Tg(NC:mCherry)*, SMCs *Tg(acta2:EGFP)*, and vascular ECs *Tg(kdrl:EGFP)*. NC-derived cells within the large chamber-like vessels (arrows). NC-derived contribution to adjacent tissues (arrowhead). Scale bars − 500 μm.

(Fig. 1f). While the SV in teleost fish has been morphologically described as a discrete vascular chamber containing "connective tissue" that lacks myocardial contribution[9,42], it has previously only been described with histological analyses[40,43–45]. Hence, the development, molecular, and cellular characteristics of the teleost SV are poorly understood and there were no apparent markers readily available that could be used to evaluate the SV in WT and *nr2f1a^aco^* mutant embryos.

To remedy the dearth of information available to examine the SV, we first conducted bulk transcriptomic analysis (RNA-seq) of the enlarged structure and single ventricle of *nr2f1a^aco^* hearts, as well as the WT SV, ventricle, atrium, and BA for comparison (Fig. 1f). Principal component analysis (PCA) and hierarchical clustering analysis of the bulk RNA-seq samples showed that the WT BA and SV had distinct expression profiles from the WT and *nr2f1a^aco^* mutant atrial and ventricular chambers and that the transcriptional profiles of the enlarged venous structure from the *nr2f1a^aco^* mutants were similar to the WT SV, but actually in between the WT BA and SV (Fig. 1g, Supplementary Fig. 4a, Supplementary Data 1 and 2). Interestingly, in comparison to each other, the WT BA and SV each had a significant number of genes with similarly enriched expression in multiple gene clusters (Fig. 1h, Supplementary Fig. 4b, and Supplementary Data 3 and 4). Moreover, many of the genes that shared expression in the WT BA and SV were markers of SMCs, including *myosin light chain kinase a (mylka), myosin heavy chain 11 a (myh11a)*, and *elastin b (elnb)*[46,47] (Fig. 1h and Supplementary Data 3 and 4).

The SMC markers were also expressed in the enlarged venous chamber-like structure from the *nr2f1a^aco^* mutants (Fig. 1h). IHC confirmed representative SMC markers were expressed in the WT BA and SV, and the enlarged chamber-like structure in *nr2f1a^aco^* mutants, yet absent from the cardiac chambers (Fig. 1i, k; Supplementary Fig. 4c). It was surprising to find the expression of SMC markers, such as Mylka and Elnb, in both the SV and BA, as they were previously reported to be specific to BA identity in teleosts and suggested to be required for patterning the outflow tract (OFT)[8,48,49]. Thus, the transcriptional profiles and IHC suggest that the WT SV shares expression of SMC markers with the WT BA and confirms the venous chamber-like structure found in the *nr2f1a^aco^* mutants is likely part of or derived from the SV.

As might be expected, many canonical markers of arterial endothelial cell (EC) identity, including *aqp8.1, spock3*, and *notch1b*[50], were enriched in the WT BA (Fig. 1h, Supplementary Data 1 and 2). However, while *nr2f1a*, which is still transcribed in *nr2f1a^aco^* mutants, was expressed in the SV, unexpectedly, many canonical EC progenitor (e.g. *etv2* and *lmo2*)[50–52] and venous EC markers (e.g. *flt4* and *lyve1b*)[53,54] were not dramatically different in the bulk RNA-seq from the WT BA and SV (Fig. 1h, Supplementary Data 1 and 2). Interestingly, some arterial EC markers that were enriched in the WT BA also showed increased expression in the *nr2f1a^aco^* mutant SV (Fig. 1h). In addition to the similar expression of SMC and venous EC markers, we found that Hox transcription factors *hoxb2a* and *hoxb3a*, were expressed in both the BA and SV, but absent from the cardiac chambers (Fig. 1h and Supplementary Fig. 4c). As much of the smooth muscle surrounding the vasculature within the anterior body is derived from neural crest[55] and neural crest (NC)-derived SMCs surround an inner vascular

endothelium of the BA that is typical of the stratified layers (tunics) surrounding large vessels[56], we asked if NC-derived SMCs also contribute to the SV. Using the combination of *Tg(sox10:GAL4,UAS:Cre)* and *Tg(-3.5ubb:loxP-EGFP-loxP-mCherry)* transgenes (together referred to as *NC:mcherry*)[56] to permanently label NC-derived cells in WT and *nr2f1a^aco^* mutant embryos, as well as examining transgenic reporters for SMCs and ECs, respectively *Tg(acta2:EGFP)* and *Tg(kdrl:EGFP)*, our data indicate that NC-derived SMCs surround an internal endothelial lining in the SV of WT and *nr2f1a^aco^* mutant fish (Fig. 1l–n, Supplementary Fig. 4d). Altogether, our analysis shows the enlarged venous structure found in the *nr2f1a^aco^* mutants is the SV, which normally is a chamber-like vascular structure adjacent to the atrium that is comprised of an endothelial lining surrounded by SMCs.

## Aberrant blood flow elicits remodeling of the SV

We next sought to understand the origin of the enlarged SV in *nr2f1a^aco^* mutants. To first determine if the *nr2f1a^aco^* mutant SV contains cells derived from embryonic atrial CMs, we performed lineage tracing with the *Tg(amhc:Cre^ERT2^)* and *Tg(ubb:LOXP-AmCyan-LOXP-ZsYellow)* (referred to as *ubb:CSY*) in WT as well as the *nr2f1a^aco^* mutant hearts (Supplementary Fig. 5a). However, cells derived from embryonic *amhc^+^* cardiomyocytes were not found distal to the atrial chamber in WT SV or within the enlarged SV of *nr2f1a^aco^* hearts (Supplementary Fig. 5b), suggesting the adaptation in *nr2f1a^aco^* mutants is intrinsic to the SV and not directly from loss of embryonic atrial CMs. To determine temporally when the SV enlarges in *nr2f1a^aco^* mutants, we examined the SMC marker Elnb in whole hearts excised at weekly intervals beginning at 7 days post-fertilization (dpf). Despite previous analysis showing Elnb is specific to the arterial pole of the zebrafish heart through 5 dpf[47], we found it also began to be expressed at the venous pole of hearts as early as 7 dpf in both WT and *nr2f1a^aco^* mutant (Fig. 2a). The size of the SV in *nr2f1a^aco^* mutants appeared to be unchanged relative to the SV in WT fish through ~21 dpf, but was discernibly enlarged in *nr2f1a^aco^* mutants compared to WT fish by 28 dpf (Fig. 2a). In addition to being enlarged by 28 dpf, IHC for Elnb on sagittal sections showed that the *nr2f1a^aco^* SV walls were significantly thickened (Fig. 2b, c). We hypothesized that wall thickening of the SV in *nr2f1a^aco^* mutants could be caused by increased cellularity, so we compared the quantity of SMCs in the WT and *nr2f1a^aco^* mutant SV. Utilizing DAPI to mark the nuclei of cells enveloped by Elnb^+^ fibers, we found there were a greater number of presumptive SMCs within the SV walls of *nr2f1a^aco^* mutants (Fig. 2d, e). Similar observations were also found in the adult SV (Supplementary Fig. 6a, b), suggesting that this thickening is not transient and likely compounds over the lifespan of the *nr2f1a^aco^* mutant fish. To determine if the increase in SV SMCs of *nr2f1a^aco^* mutants was due to increased proliferation, juvenile fish were pulsed with EdU at 21 and 24 dpf via pericardial injection (Fig. 2f). Indeed, we found that there was a significant increase in the percentage of EdU^+^-Elnb surrounded cells within the SV, the majority of which were concentrated in regions undergoing thickening (Fig. 2h, i). Analysis of immune cells did not show infiltration that could be contributing to the thickening in the *nr2f1a^aco^* mutants. Thus, our data show that the SV of *nr2f1a^aco^* mutants

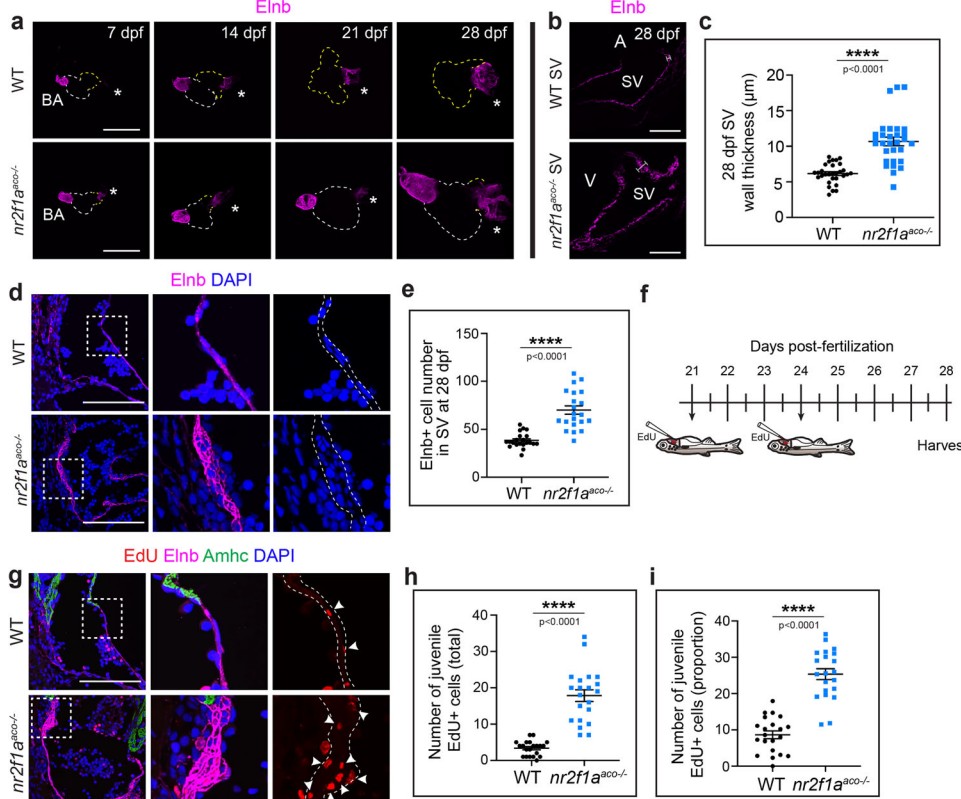

**Fig. 2 | *Nr2f1a* mutant SV undergoes adaptive remodeling via proliferation of smooth muscle cells. a** Confocal images of Elnb IHC in whole hearts from WT and *nr2f1a^aco^* fish at weekly intervals through 28 dpf. Venous pole (asterisks). Dashed lines outline the myocardium (white - ventricle, yellow - atrium). WT 21 dpf and 28 dpf images just show the atrium and venous pole. Scale bars – 200 µm. **b** Confocal images of IHC for Elnb on mid-sagittal sections showing the SV at 28 dpf. Bars indicate thickness. Scale bars – 100 µm. **c** Quantification of SV wall thickness (µm) in sections (*n* = 10 fish). Individual points on the graphs represent a single section averaged from 10 measurements from a single section (3 total sections per fish). 3 adjacent sections were analyzed per fish. **d** Confocal images of IHC for Elnb (magenta) and DAPI (blue) staining on sagittal sections showing the SV at 28 dpf. DAPI⁺ nuclei within the Elnb staining of the whole sections were counted. Boxes show the regions of the higher magnification. Dashed outline indicates the Elnb⁺ region. Scale bars – 50 µm. **e** Quantification of Elnb-surrounded cells (DAPI⁺ nuclei)

within sections of the SV (*n* = 7 fish per group). 3 adjacent sections were quantified per fish. **f** Schematic of EdU labeling with intrathoracic injections in juvenile zebrafish. **g** Confocal images of sagittal sections from juvenile WT and *nr2f1a^aco^* mutant fish showing the SV following detection of EdU (red) incorporation. Boxes indicate regions of higher magnification. Dashed outline indicates DAPI⁺ nuclei (blue) within the Elnb⁺ (magenta) tissue. Amhc (green) was used as control to mark the border of the SV. White arrowheads indicate EdU⁺ nuclei embedded in the Elnb⁺ tissue of the SV. Scale bars – 50 µm. **h, i** Quantification of a total number of Elnb-surrounded EdU⁺ nuclei and the proportion of Elnb-surrounded EdU⁺ to total (DAPI⁺) nuclei within sections SV (*n* = 7 fish per group). Individual points represent 1 section with 3 sections counted per fish. An unpaired, two-sided Student's *t*-test was used to assess statistical differences between WT and *nr2f1a^aco^* in all graphs. Comparisons marked ****p < 0.0001. Error bars represent the mean +/− SEM. Source data are provided as a Source Data file.

undergoes a remarkable remodeling response that begins around 21 dpf and entails thickening of the wall due to increased SMC cellularity from proliferation.

Vascular remodeling and hyperplasia of SMCs, such as occur with diseases like pulmonary hypertension[57], can be caused by improper blood flow[39]. To explore if aberrant blood flow occurs in *nr2f1a^aco^* mutant hearts, potentially increasing shear stress within the SV, we performed echocardiography on adult WT and *nr2f1a^aco^* mutant fish[58]. Retrograde blood flow was apparent from the ventricle into the SV of *nr2f1a^aco^* mutants, which was never observed in the hearts of WT control fish (Fig. 3a, Supplementary Movies 1 and 2). This observation led us to posit that regurgitation in the *nr2f1a^aco^* mutant hearts could be increasing shear stress within the SV and contributing to the thickening phenotype. Accordingly, we sought to decrease the mechanical stress imposed by these aberrant hemodynamic forces within the SV. Pharmacological vasodilators Captopril and Enalapril were chosen as mitigating agents because they are used clinically to treat hypertension[59,60], while estradiol was chosen for its association with improving pulmonary hypertension in animal models[61]. WT and *nr2f1a^aco^* mutant fish were treated with 250 µM Captopril or DMSO (vehicle control) beginning at 14 dpf until 28 dpf. Although WT fish

were unaffected, *nr2f1a^aco^* juvenile mutants exposed to Captopril showed a significant reduction in the thickness of the Elnb⁺ SV walls compared to the control treatments (Fig. 3b, c). We also found that fish were amenable to longer-term treatments with Enalapril and estradiol. WT and *nr2f1a^aco^* mutant fish were treated beginning at 48 hpf with 50 µM Enalapril or ethanol (vehicle control) and 10 nM estradiol or DMSO (vehicle control) until 28 dpf. With these prolonged treatments, again *nr2f1a^aco^* mutants exposed to Enalapril or estradiol both showed a significant reduction in the thickness of the Elnb⁺ SV walls compared to the WT control fish (Supplementary Fig. 7a–d). Conversely, we asked if increasing blood flow, and therefore sheer stress, could exacerbate the SV thickening in *nr2f1a^aco^* mutant fish using a swim tunnel, an established exercise model[62]. *nr2f1a^aco^* fish were exercised in a swim tunnel via pacing beginning at 21 dpf daily for 30 min over 2 weeks (Fig. 3d, Supplementary Movie 3). Indeed, we found that this exercise regimen was able to elicit an increase in SV wall thickness in juvenile *nr2f1a^aco^* mutants compared to unexercised control sibling *nr2f1a^aco^* mutant fish (Fig. 3e, f). Surprisingly, despite the enlargement of the ventricle, lack of atrium, and adaptive thickening of the SV, adult *nr2f1a^aco^* mutant fish performed as well as WT fish in sprint and endurance trials (Supplementary Fig. 8a, b), suggesting that their

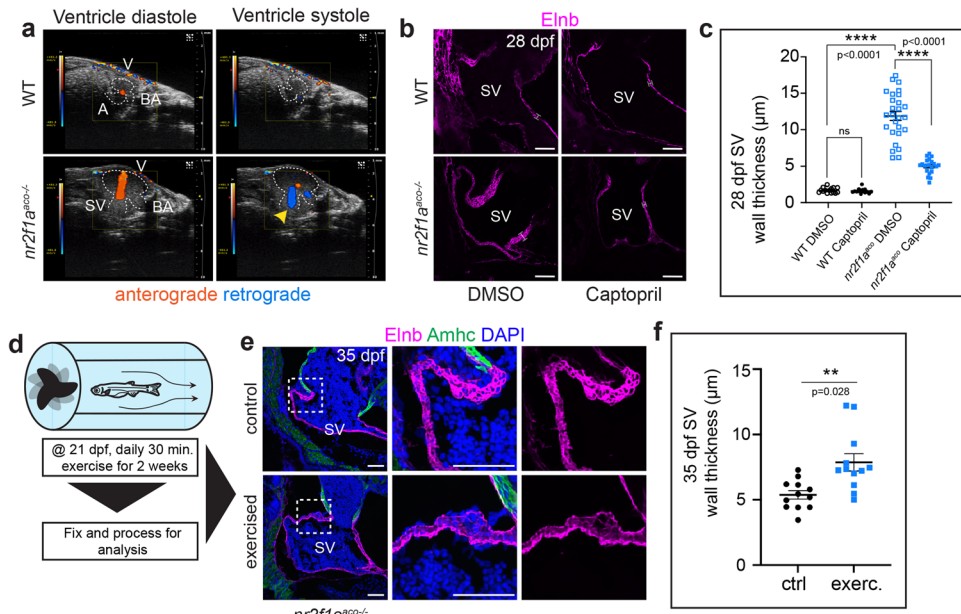

**Fig. 3 | Aberrant blood flow contributes to remodeling in *nr2f1a* mutant hearts.**
**a** Ultrasound images from echocardiography were conducted on adult WT and *nr2f1a*$^{aco}$ mutant fish. Anterior is to the right and ventral is up. Flow direction was pseudo-colored for ease of clearly discerning anterograde ventricular inflow (orange) and retrograde outflow (blue). Dashed lines denote the borders of heart. Yellow arrowhead indicates the location of regurgitation in *nr2f1a*$^{aco}$ mutant heart. Still images were isolated from Supplementary Movies 1 and 2. *n* = 3 WT and *nr2f1a*$^{aco}$ mutant fish were examined. **b** Confocal images of sagittal sections from 28 dpf WT and *nr2f1a*$^{aco}$ mutant fish treated with either Captopril or DMSO (control). Scale bars – 100 μm. **c** Quantification of Elnb$^+$ wall thickness (μm) at 28 dpf following Captopril treatments began at 14 dpf. Individual points on the graphs represent 10 measurements averaged from different regions of the SV in a single section (3 total sections per fish). (*n* = 5 for WT control and treated groups, *n* = 9 for *nr2f1a*$^{aco}$ control group, *n* = 7 for *nr2f1a*$^{aco}$ treated group). Statistical significance was

calculated using an ordinary one-way ANOVA with multiple comparisons. Comparisons marked ****$p$ < 0.0001. Source data are provided as a Source Data file. **d** Schematic outlining the juvenile exercise regimen beginning at 21 dpf. **e** Confocal images of sagittal sections showing immunostaining detecting Elnb (magenta), Amhc (green), and DAPI$^+$ nuclei (blue) in the SV of unexercised (control) and exercised *nr2f1a*$^{aco}$ mutant fish following 2 weeks of daily exercise. Scale bars – 50 μm. **f** Quantification of Elnb$^+$ wall thickness at 35 dpf following 2 weeks of daily exercise (*n* = 4 fish per group). Individual points on the graphs represent 20 measurements averaged from different regions of the SV in a single section (3 sections per fish). An unpaired, two-sided Student's *t*-test was used to assess statistical differences between exercised and unexercised *nr2f1a*$^{aco}$ fish. Comparison marked **$p$ = 0.0028. All error bars represent the mean +/– SEM. Source data are provided as a Source Data file.

---

hearts are functionally able to compensate. Together, these findings support the idea that the SV of *nr2f1a*$^{aco}$ mutants undergoes adaptive remodeling influenced by shear stress and improper regurgitative blood flow.

**Heterogenous SMC and EC populations comprise the BA and SV**
We next sought to gain an understanding of the specific cell types that normally comprise the zebrafish BA and SV and how they are affected in the adapting SV of the *nr2f1a*$^{aco}$ mutants. Hence, we performed scRNA-seq on the BA and SV of WT and the *nr2f1a*$^{aco}$ mutant fish (Fig. 4a). Unsupervised joint-clustering (ICGS2) of BA and SV of adult WT and *nr2f1a*$^{aco}$ mutants identified 28 cell populations defined by unique marker genes, with all clusters containing both WT and *nr2f1a*$^{aco}$ mutant cells (Supplementary Fig. 9a, b and Supplementary Data 5). Manual annotation based on established markers found that 13 of these clusters represent contaminant immune cells, atrial CMs, and hepatocyte-like cells. Thus, we focused on the remaining 15 clusters from our analysis, as they reflected the primary cell types intrinsic to both chambers. UMAP visualization of these primary BA and SV populations found that cells separate primarily by genotype and capture, even within each annotated cell-population (Fig. 4b, c). Nonetheless, these cell population predictions were consistent with those obtained with supervised alignment to only WT captures (Supplementary Fig. 10a–c). Annotation of the cell clusters based on marker gene expression from the WT and *nr2f1a*$^{aco}$ mutant BA and SV indicated the major cell populations were predominantly SMCs and ECs (Fig. 4b, Supplementary Fig. 11 and Supplementary Data 6). Specifically, clusters (C) 1–8 were identified as being SMCs, with C1-3 and C6 expressing

high levels of canonical SMC markers, including *myh11a, elnb, mylka,* and *acta2* (Fig. 4d, Supplementary Fig. 12)[46]. C4, C5, C7, C8 expressed lower levels of these canonical SMC markers but were enriched for fibronectins (Fig. 4d, Supplementary Fig. 12). While the interpretation of UMAP embeddings can be problematic, these plots suggest that overall SMC (C3-8) and EC (C9-12) populations in each the BA and SVs of the WT and *nr2f1a*$^{aco}$ mutant tissues are more proximal, with the exception of the main *nr2f1a*$^{aco}$ mutant and WT BA SMCs clusters C1 and C2, which display greater separation than these other populations (Fig. 4b, c). The multiple clusters of SMCs in the BA (C1-3) and SV (C4,6,7,8) of both conditions support there is heterogeneity within these populations at both poles of the heart (Fig. 4b–d).

Similar to the SMC populations, ECs within the WT and *nr2f1a*$^{aco}$ mutant BA and SV primarily segregated within the UMAP based on their arterial (C10) or venous (C9) tissue of origin irrespective of if they were from WT or *nr2f1a*$^{aco}$ mutants (Fig. 4b–d and Supplementary Fig. 13). Manual annotation based on enrichment for previous established EC markers[63–65], such as *flt1, spock3,* and *aqp8.1* (arterial), and *lyve1b* and *stab2* (venous)[50] support that C10 and C9 are respectively arterial and venous-lymphatic ECs (Fig. 4d, Supplementary Data 6). Furthermore, gene-set enrichment (GO-Elite) analysis comparing these clusters to previously reported scRNA-seq analysis of 24 hpf embryonic ECs[50] affirmed that the WT C10 and C9 ECs, respectively, shared expression of established arterial and venous markers from the embryonic vasculature (Fig. 4e and Supplementary Data 7). The previous scRNA-seq analysis also suggested that there are ECs of intermediate transcriptional signature located in the cranial vasculature, as they express both canonical arterial and

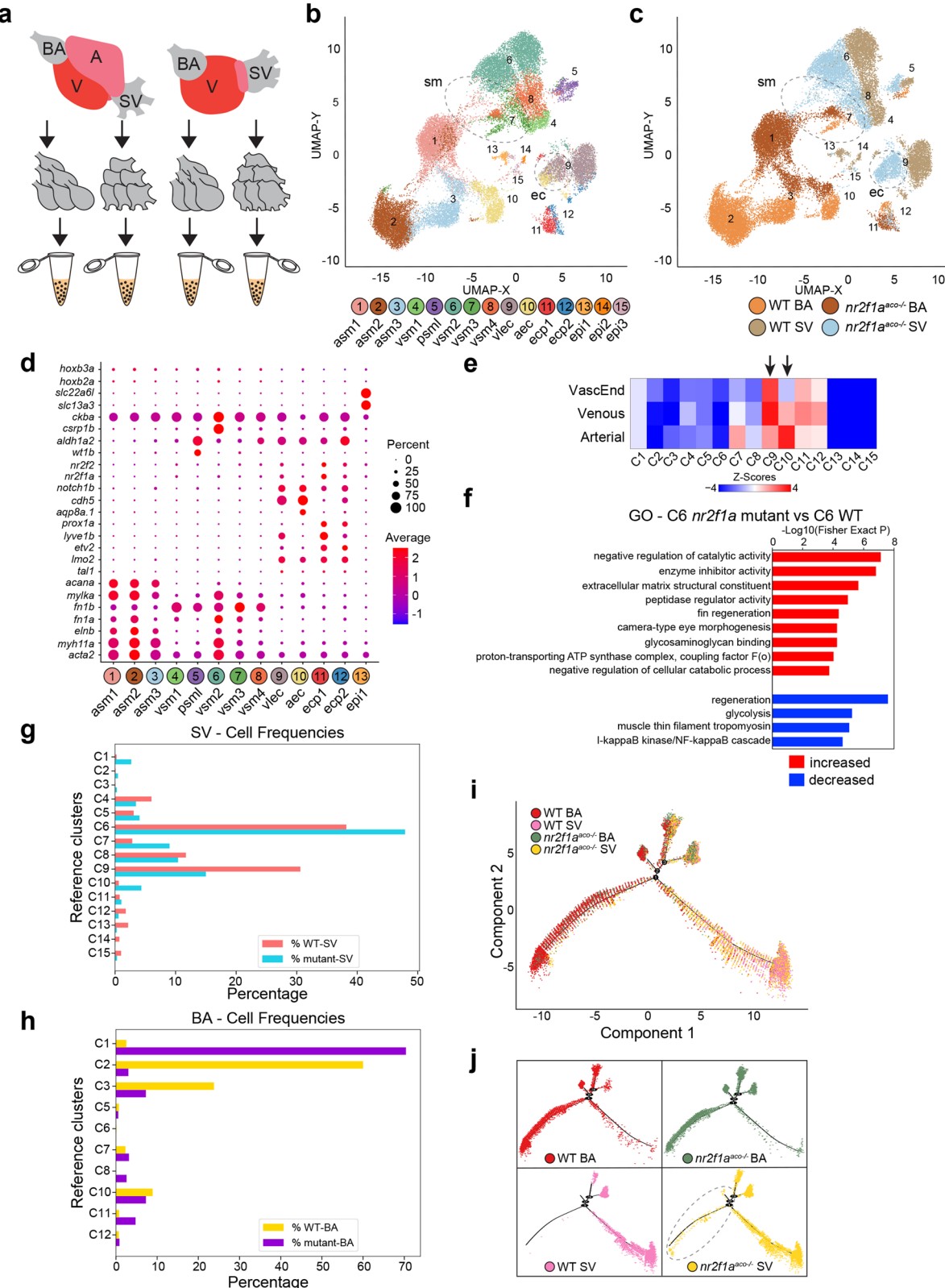

venous markers[50]. Interestingly, gene-set enrichment analysis with WT BA and SV clusters also suggested that the venous WT C9 shared enrichment of markers with this intermediate vascular EC population (Fig. 4e and Supplementary Data 7). Consistent with venous ECs of the WT SV having an identity that may be intermediate to other arterial and venous EC populations, genes, including *notch1b* and *cdh5*, which are typically enriched in arterial ECs[50,66], were also

expressed in the WT and *nr2f1a^{aco}* C9 ECs (Fig. 4d and Supplementary Fig. 13). In addition to these larger EC populations, C11 and C12 were smaller EC populations that were found in both the BA and SV of the WT and *nr2f1a^{aco}* mutant embryos that co-expressed canonical EC progenitor markers, including *etv2*, *tal1*, *lmo2*, and *prox1a*[50,65], as well as the aforementioned arterial and venous markers (Fig. 4d and Supplementary Fig. 14). Hence, in addition to the larger arterial and

**Fig. 4 | The remodeled *nr2f1a* mutant SV acquires characteristics of the arterial BA. a** Schematic depicting the dissection of individual cardiac chambers for single cell RNA-seq analysis. **b** Uniform Manifold Approximation and Projection (UMAP) plot showing the 15 ICGS2 clusters corresponding to smooth muscle, endothelial, and epithelial cells. asm – arterial smooth muscle, vsm – venous smooth muscle, psml – pan-smooth muscle-like, vlec – venous-lymphatic endothelial cell, aec – arterial endothelial cell, ecp – endothelial cell progenitor, epi – epithelial cell. **c** The UMAP plot shows the contributions of cells from each the WT BA, WT SV, *nr2f1a^aco* mutant BA, and *nr2f1a^aco* mutant SV. Dashed outlines indicate cells from clusters 1, 7, and 10 (SMCs) and clusters 9 and 10 (ECs) occupying similar space that are from the WT BA, WT SV, and *nr2f1a^aco* mutant SV. **d** Dot matrix plot comparing the expression of selected SMC and ECs marker genes within clusters 1–13. **e** Gene-enrichment (GO-Elite) of WT cluster marker genes compared to previously published arterial, venous, and intermediate vascular populations. **f** GO enrichment of marker genes from *nr2f1a^aco* mutant C6 vs WT C6 SMC populations. Gene-set enrichment analysis is performed using GO-Elite. A Z-score and Fisher's Exact two-sided Test p-value are calculated to assess over-representation of Ontology terms, gene-sets, and pathways. Adjusted p-values (FDR p-values) for these various tests are calculated using the Benjamini-Hochberg correction method. **g** Frequency of cells from clusters in the *nr2f1a^aco* mutant and WT SV. **h** Frequency of cells from clusters in the *nr2f1a^aco* mutant and WT BA. **i** Pseudotime analysis was performed with Monocle2 mapping cellular identities from the SMC clusters (1–8). **j** Individual 10x Genomics captures showing separated populations from (**i**). Dashed outline indicates *nr2f1a^aco* mutant SV cells occupying trajectories that are normally found almost exclusively in the WT and *nr2f1a^aco* mutant BA.

venous EC populations, smaller putative EC progenitor cells are also found in both the BA and SV.

## Arterialization of SMCs and ECs in the adapting SV

We next evaluated changes in gene expression in the *nr2f1a^aco* mutant SV cell populations compared to the WT. We were particularly interested in determining if arterial-venous fate transformations were occurring in the adapting venous tissues of the *nr2f1a^aco* mutant SV, since Nr2fs are associated with promoting venous EC identity in mammals[53], and because in addition to the morphology of the *nr2f1a^aco* mutant SV becoming thicker, which was reminiscent of the BA, PCA of our bulk RNA-seq suggested that the *nr2f1a^aco* mutant SV becomes molecularly more similar to the WT BA (Fig. 1g, h and Supplementary Fig. 4a). Using the cellHarmony software, differentially expressed genes (DEGs) were identified in each of the individual cell clusters from the WT and *nr2f1a^aco* mutant SV (Supplementary Fig. 15; Supplementary Data 8 and 9). With respect to DEGs whose expression was increased specifically in C6 cells (the major venous SMC population (Fig. 4b–d)) when the *nr2f1a^aco* mutant SV was compared to WT SV, Gene Ontology (GO) terms were associated with extracellular matrix composition and muscle contraction (Fig. 4f and Supplementary Data 10). Specifically, collagens (*col1a1a* and *col1a1b*) and *elnb* were among the extracellular components found to have increased expression in the *nr2f1a^aco* mutant C6 SMCs relative to the WT C6 SMCs (Supplementary Data 10). These findings are consistent with increased ECM deposition and changes within the ECM associated with the morphological thickening exhibited by the SV in *nr2f1a^aco* mutants. Overall, GO terms in most of the *nr2f1a^aco* mutant SV cell populations that were increased relative to WT SV cells were associated with translation (translation - GO:0006412, translational elongation - GO:0006414, translation elongation factor activity - GO:0003746) and regeneration (regeneration - GO:0031099, fin regeneration - GO:0031101), while those that were decreased were associated with muscle contraction (muscle contraction - GO:0006936, muscle thin filament tropomyosin - GO:0005862) and metabolism (mitochondrial electron transport, cytochrome c to oxygen - GO:0006123, respiratory electron transport chain - GO:0022904) (Supplementary Fig. 16a; Supplementary Data 11), which is consistent with cardiovascular stress and remodeling observed in the *nr2f1a^aco* mutant SV.

To address potential changes in cell fates associated with adaptive remodeling, we examined the frequency of the different SMC and ECs populations within the WT and *nr2f1a^aco* mutant SV. The *nr2f1a^aco* mutant SV had SMCs assigned to C1, which are normally exclusively found in the WT and *nr2f1a^aco* mutant BA, and an increase in C7 SMCs, a population found in both the WT and *nr2f1a^aco* mutant BA (Fig. 4g). Within the UMAPs, although the majority of *nr2f1a^aco* mutant SV SMC populations were still located adjacent to the WT SV populations, importantly, the C1 and C7 SMCs from the *nr2f1a^aco* mutant SV were found to be located either with or intermediate to *nr2f1a^aco* mutant and WT BA-derived populations (Fig. 4b, c), suggesting they are acquiring more arterial-like identity. To further assess the similarity of

differentiation states within the BA and SV SMC populations (C1-8), pseudo-temporal ordering of the transcriptional profiles was performed using the software Monocle[67]. This analysis showed that in contrast to the SMCs from the WT SV, SMCs from the *nr2f1a^aco* mutant SV had a similar differentiation state to arterial SMCs from both the WT and *nr2f1a^aco* mutant BA (Fig. 4i, j), again supporting a more arterial-like transcriptional profile in these SMCs. Additionally, a comparison of DEGs from the *nr2f1a^aco* mutant SV clusters to WT BA and SV marker genes showed an enrichment of *nr2f1a^aco* mutant C6 markers with WT C2, the major WT BA SMC population (Supplementary Fig. 16a, b; Supplementary Data 12–14). Thus, our data indicate that subpopulations of SMCs from the *nr2f1a^aco* mutant SV appear to have acquired more arterial SMC identity.

With respect to EC identities in the *nr2f1a^aco* mutant SV, the frequency of arterial ECs in C10, which are almost exclusively derived from the WT and *nr2f1a^aco* mutant BA, was dramatically increased (Fig. 4g). Furthermore, within the UMAP, the C10 ECs from the *nr2f1a^aco* mutant SV more frequently intermix with the *nr2f1a^aco* mutant venous-lymphatic ECs of C9 (Fig. 4c), suggesting they are arterial-like ECs yet still have intermediate identity to the *nr2f1a^aco* mutant and WT C9 venous-lymphatic ECs. Identification of unique markers from the WT and *nr2f1a^aco* mutant arterial (C10) and venous-lymphatic (C9) populations confirmed that the C10 ECs derived from the *nr2f1a^aco* mutant SV share lineage defining markers with the WT arterial BA-derived C10 ECs (Supplementary Fig. 17a). Analyzing the differentiation states of C9 and C10 populations by lineage pseudotime analysis (SlingShot) showed that the *nr2f1a^aco* mutant C10 ECs were closer in their transcriptional state to the BA-derived arterial C10 ECs, but still intermediate to most WT and *nr2f1a^aco* mutant SV-derived C9 ECs (Supplementary Fig. 17b). Thus, similar to SMCs, subpopulations of ECs from the *nr2f1a^aco* mutant SVs also appear to have a more arterial identity like ECs from the WT and *nr2f1a^aco* mutant BAs.

*Nr2f1a* expression is restricted to atrial/venous tissues of the embryonic[38] and adult hearts (Supplementary Fig. 18). Additionally, the scRNA-seq analysis showed that it, as well as *nr2f2*, are predominantly expressed within both the venous SMC and EC clusters, but not appreciably in cell clusters from the BA, except for the EC progenitor clusters (C11 and C12) (Fig. 4d, Supplementary Fig. 13). Therefore, it was interesting that one of the changes observed from the scRNA-seq analysis was between the major SMC populations found in the WT and *nr2f1a^aco* mutant BA. C2 arterial SMCs were almost exclusively from the WT BA, while C1 arterial SMCs were almost exclusively from the *nr2f1a^aco* mutant BA (Fig. 4h). Additionally, the *nr2f1a^aco* mutant C1 population appears intermediate between the WT BA and *nr2f1a^aco* mutant SV SMCs via conventional UMAP analysis (Fig. 4b). More broadly, examining DEGs in the *nr2f1a^aco* mutant BA clusters compared to the WT BA and SV marker genes combined showed that genes with increased expression in the *nr2f1a^aco* mutant BA were enriched in WT BA lineage markers (Supplementary Fig. 16c, d and Supplementary Data 15–17), similar to the WT and *nr2f1a^aco* mutant SV. Specifically, comparing the C1 and C2 populations respectively from

the $nr2f1a^{aco}$ mutant and WT BA, GO terms associated with DEGs decreased in C1 relative to C2 were also associated with muscle contraction and metabolism, while increased terms were associated with translation and regeneration (Supplementary Data 18 and 19). Collectively, our work shows that the adaptation of the SV in zebrafish $nr2f1a^{aco}$ mutants from aberrant hemodynamic forces is associated with increased SMC proliferation, changes in cellular metabolism and growth, and a shift in the molecular identity of some venous SMCs and ECs subpopulations to arterial-like identities that are found with the BA.

### Equivalency of the *Ciona* blood sinuses

The adult $nr2f1a^{aco}$ mutant hearts, which effectively lack an atrial chamber and whereby the SV thickens and acquires arterial, BA-like characteristics adjacent to a single ventricle, attain a form that is reminiscent of the single-chamber hearts with flanking BSs found in tunicates[68]. While there have been histological analyses[2,68], molecular characteristics of the adult tunicate BSs are lacking. Thus, we postulated that the tunicate heart and flanking BSs, which we have designated the arterial pharyngeal sinus (PS) and venous stomach sinus (SS) according to their anatomical proximity to the pharynx and stomach,

respectively, may reflect an ancestral cardiac ground state with the greater equivalency of cell populations on both sides of the heart, as blood can be expelled bidirectionally[69] (Fig. 5a). To investigate if the molecular characteristics of the tunicate BSs provide evidence of ancestral BA-SV symmetry, we compared gene expression from bulk RNA-seq of the central tubular heart to the lateral BSs from adult *Ciona robusta* (Fig. 5a and Supplementary Fig. 19a), mirroring the bulk RNA-seq analysis of the WT adult zebrafish cardiac chambers, BA, and SV described above. Orthologs of canonical myocardial genes, including *tropomyosin* (*tpm*), *troponin I* (*tnni*), and the transcription factor *nk4* (*nkx2.5* ortholog) were found enriched in the central portion of the heart (Fig. 5a, b, Supplementary Fig. 19b, and Supplementary Data 20 and 21). However, *Ciona* transcription factors *hox3*, *pdx1*, *six1/2*, *isl*, *fli/erg*, and *ets/pointed*, and the mechanosensitive channel *piezo* were found to have enriched expression in both the PS and SS (Fig. 5a,b, Supplementary Fig. 19b, and Supplementary Data 20 and 21). Furthermore, a previous report corroborates *hox3* expression in tissues flanking the *Ciona* heart[70]. Genes associated with the endostyle, a mucus producing organ in the pharynx of tunicates, were found in the PS (endostyle specific genes 1–3 (endostyle spg1-3)), while genes associated with digestion were found in the SS (gh18 and ciap) (Fig. 5b).

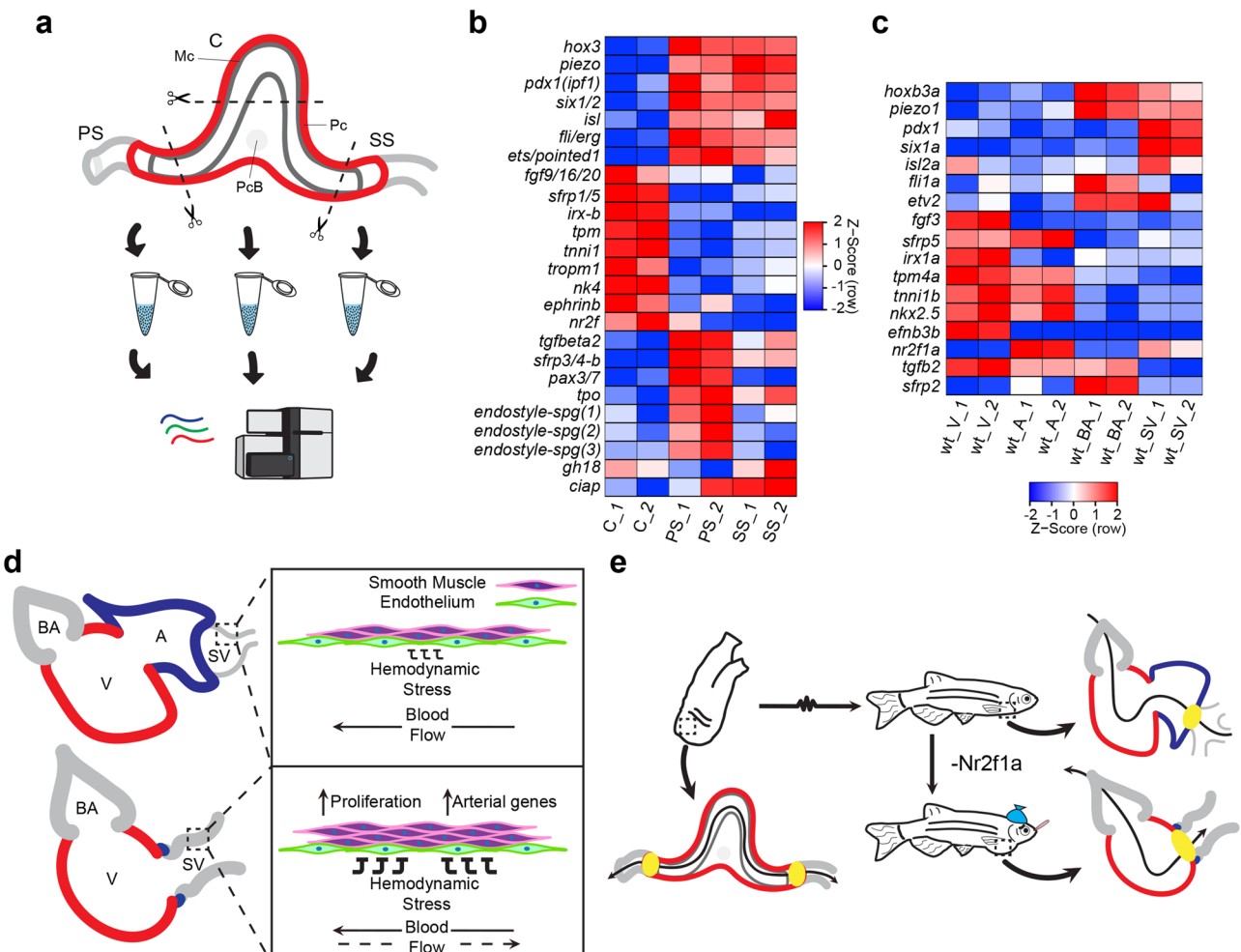

**Fig. 5 | Blood sinuses of the tunicate heart have conserved expression of genes found in the teleost BA and SV. a** Schematic summarizing the regions of the adult *Ciona* heart that were harvested for bulk RNA-seq. Pharyngeal blood sinus (PS), stomach blood sinus (SS), and cardiac tube (C). Mc – myocardial tissue (dark gray), Pc – pericardium (red), PcB – pericardial body. **b** Heatmap showing the expression of representative genes found in the C, PS, and SS of the adult *Ciona* heart. **c** Heatmap showing the expression of homologous genes to those in *Ciona* found in the V, A, BA, and SV of WT adult zebrafish hearts. **d** Summary depicting the adaptive response whereby the SV thickens due to proliferation from SMCs in $nr2f1a^{aco}$ mutants. **e** Evolutionary model depicting the heart of the *Ciona* with bi-direction flow and the adaptive response in $nr2f1a^{aco}$ mutants to regurgitation whereby the heart comes to resemble characteristics of the *Ciona* heart with equivalency in the sinuses at the arterial and venous poles.

While these likely reflect some tissue contamination from the dissections, they serve to validate the anatomical origin of the PS and SS tissue samples. Zebrafish orthologs of *Ciona* genes found in both the PS and SS were found to be enriched in both the BA and SV (*hoxb3a* and *piezo1*), while some zebrafish orthologs, or at least homologous genes, showed enriched expression in either the BA (*fli1a* and *etv2*) or SV (*pdx1*, *six1a*, *isl2a*) (Fig. 5c). Furthermore, the scRNA-seq analysis indicated that zebrafish *hoxb3a*, *six1a*, *pdx1*, and *isl2a* are enriched in SMC populations, *piezo1* is enriched in SMC and EC populations, and confirmed *fli1a* and *etv2* are enriched in EC populations (Fig. 4d). Although our analysis of the *Ciona* BSs supports our hypothesis that there is greater equivalency of these vascular tissues, it is worth noting that some asymmetry in gene expression was also identified. With respect to the cardiac tissue, genes associated with controlling development of the arterial pole of vertebrates from the later differentiating SHF, including *tgfb2*/the TGFβ signaling pathway[11], were found in the PS of the *Ciona* heart (Fig. 5b), as well as the zebrafish BA (Fig. 5c). Similar to the polarization of gene expression found with homologs of the *Ciona* genes in the zebrafish BA and SV, some homologs of genes found in the single *Ciona* cardiac chamber (*irx-b*, *fgf9/16/20*, *ephrinb*, and *nr2f*) showed polarized expression in either the ventricle or atrium of the zebrafish. Specifically, zebrafish *irx1a*, *fgf3*, and *efnb3b* were enriched in the ventricle, while *nr2f1a* is enriched in atrial CMs (Fig. 5c). Thus, comparison of gene expression profiles of adult *Ciona* BSs with BA and SV from zebrafish support a hypothesis that the more specialized BA and SV in vertebrates may have originated from BSs flanking an ancestral chordate heart that exhibit greater equivalency (Fig. 5a).

## Discussion

### A model for adult patients with structural CHDs

Our previous work demonstrated that embryonic *nr2f1a* mutants have smaller atria due to reduced AC differentiation[38]. Here, we show that despite these embryonic atrial defects, zebrafish *nr2f1a*<sup>aco</sup> mutants can survive to adulthood effectively without an atrial chamber. Invariably, the surviving fish develop secondary complications reminiscent to those found in human patients with ASDs, including sinus venosus ASD[71]. ASDs are more commonly observed in adult patients with CHDs[24]. Prolonged cardiac dysfunction in humans, such as can occur in adult patients with ASDs, can impart sustained stress on arterial and venous vessels proximal to the heart due to improper blood flow[39,57], with pulmonary hypertension (PH) being one of the secondary complications that can arise later in life[72]. We observe that the hearts of the adult *nr2f1a* mutants have enlarged ventricles and SV, suggestive of sustained hemodynamic stress to these hearts. Our data support that aberrant blood flow and regurgitation occur between the ventricle and SV of the *nr2f1a* mutants, as the fish effectively lack an atrial chamber. We posit the adaptive remodeling of the SV is due to prolonged aberrant flow and shear stress within this venous chamber, again similar to what is seen in adult patients with ASDs[73–75]. Supporting the hypothesis that the adaptation of the SV in *nr2f1a* mutants is a vascular hypertensive response sharing conserved mechanisms that underlie PH in humans, we find that more acute or prolonged treatments of the *nr2f1a* mutants with vasodilators, including ACE inhibitors[76,77] and estradiol[61,78,79] are able to mitigate the SV thickening in *nr2f1a* mutants. Moreover, while numerous mammalian and zebrafish models of CHDs and PH exist[80–82], the zebrafish *nr2f1a* mutants are unique in that they parallel the prolonged pathology of adult patients with CHDs by initiating with embryonic structural heart defects and the progression to later secondary vascular remodeling in surviving juveniles and adults. Therefore, continued investigation of the cardiovascular defects in adult zebrafish *nr2f1a*<sup>aco</sup> mutants has the potential to provide novel insights into the pathology of adult patients suffering from CHDs and could open doors to new therapeutics.

## Composition of the teleost SV

Our work also fills longstanding gaps in our understanding of the molecular and cellular characteristics of the teleost SV. The SV as an anatomical structure was first described decades ago in fishes[83,84], as well as in the earlier branching agnathans[85]. However, it has been a largely understudied component of these vertebrates' hearts. Previous studies in fish and agnathans have been exclusively histological[86–88], with the SV in teleosts being described as consisting of "connective tissue"[40]. Although histological studies of agnathans have stated its SV is comprised of smooth and some cardiac muscle[9,87,88], the specific cell types have not been verified with established molecular markers. Our transcriptomic (bulk and scRNA-seq) coupled with verification using IHC analysis supports that the SV in zebrafish is comprised of an outer layer of SMCs, with canonical markers such as Elnb and Mylka[8,47–49], appearing at approximately 7 dpf. Previous reports had suggested smooth muscle was specific to the arterial pole of the zebrafish heart. However, these studies failed to investigate potential SV-localized expression past 5 dpf[47,48]. Our genetic lineage tracing supports that some of the SMCs in the SV are neural crest-derived, similar to the SMCs found in the BA[55] and consistent with the cranial vasculature[56]. Our single cell transcriptomic analysis identified heterogeneous SMC populations in the BA and the SV, providing an atlas of the specific cell populations within these auxiliary chambers and contributing to recent analysis of SMC types associated with the arterial and venous vascular tissues in vertebrate hearts[89,90]. One of the main identifiers of the different SMC populations with the zebrafish BA and SV is the expression of *fibronectin* (*fn*) genes, with Fn being associated with SMCs showing greater proliferation in mammals[91,92]. Future analysis will be aimed at interrogating the functional significance of these different SMC populations with the BA and SV and their contributions to the hypertensive response of the SV.

The other major cell type identified in the single cell analysis was ECs, which line the lumen of both the BA and SV. Arterial and venous EC populations were clearly distinguished from the BA and SV of WT and *nr2f1a*<sup>aco</sup> mutant fish via the expression of established marker genes and correlated with the expected tissue of origin in the scRNA-seq UMAP. In addition to these larger EC populations that segregated with the BA and SV, there were smaller clusters of EC progenitors from both the BA and SV of WT and *nr2f1a*<sup>aco</sup> mutant fish that co-expressed established arterial and venous markers genes. Despite the distinction of larger arterial and venous EC populations within these auxiliary chambers, it was interesting that many markers of the arterial EC identity were also expressed in the venous ECs from the WT SV. This less distinct expression of arterial and venous markers in the WT SV is reminiscent of recent single cell transcriptomic analysis of ECs in embryonic zebrafish, which identified a population of cranial vascular ECs that were of intermediate identity that co-expressed canonical arterial and venous markers[50]. Gene-enrichment analysis comparing gene expression of the embryonic cranial vascular ECs and venous ECs from the WT SV supported that the ECs in the WT SV shared gene expression with these embryonic ECs of intermediate identity. Furthermore, the embryonic intermediate identity of these EC populations in these embryos was associated with more pressure sensing vasculature[50]. Hence, it is logical that the ECs of the zebrafish SV, which is necessary to receive systemic blood prior to entering the heart, and as our data show undergo adaptive thickening in response to aberrant hemodynamic forces, would be suited to sense changes in blood pressure.

Despite the dramatic morphological changes in the SV of the adult *nr2f1a* mutants, the single cell transcriptomics showed that the fundamental cell populations identified from the BA and SVs of WT and *nr2f1a* mutants were largely similar. However, there were SMC and EC subpopulations from the *nr2f1a* mutant SVs that were assigned to the arterial SMCs (C1 and C7) and arterial ECs (C10) based on their

transcriptional signatures, supporting the hypothesis that some of these SMCs and ECs are taking on arterial identity. *Nr2f1a* was consistently associated with venous identity in the bulk and scRNA-seq data sets, being expressed in both the venous SMC and EC populations. In addition to promoting atrial CM differentiation[28], Nr2f2 in mammals is required to promote venous EC identity through repressing Notch signaling[93]. We are not aware of previous work demonstrating that *Nr2f* genes are required to perform similar roles in repressing arterial SMC identity in venous SMC. Our data would suggest this is a possibility that will require further analysis. Nevertheless, it is important to emphasize that it is only a subset of SMC and EC populations that had arterial identity and not a complete transformation of these populations, as may have been predicted from some mammalian studies of Nr2f-dependent arterial-venous specification[93–95]. Although we have not yet performed single cell transcriptomic analysis of *nr2f1a* mutant vasculature in embryos, we have not observed discernible changes in the identity of major vessels (the dorsal aorta and posterior cardinal vein) in embryos (Supplementary Fig. 20a–c). Moreover, given that we observe the SV defects initiate in the juveniles at ~3 weeks post-fertilization, we presently cannot distinguish if these transformations are directly due to the loss of *nr2f1a* within these cells promoting/maintaining venous SMC and EC identity at later stages or they are secondary consequences due to the aberrant hemodynamic forces. It is also worth noting that there is no indication that Notch signaling is affected by any of our sequencing data (Supplementary Fig. 15 and Supplementary Data 13, 14, 16, 17). Because *nr2f1a* is not expressed in cells other than a small population of EC progenitors in the BA, it is also interesting that the SMC populations from the BA of *nr2f1a* mutants within our UMAP data are actually located in between the *nr2f1a* mutant SV and the WT BA. Furthermore, we did not observe arterial-venous fate transformations within BA-derived subpopulations. However, the most prevalent DEGs and GO terms were associated with vascular contractility, metabolic changes, and stress markers. Hence, the gene expression changes are consistent with the *nr2f1a* mutant BA having reduced anterograde blood flow and shear stress, which is why it appears to be more venous-like based on its location in the UMAP relative to the SMC in the WT BA. Altogether, our data support that only subpopulations of SMCs and ECs within the adapting *nr2f1a^aco* mutant SV are taking on arterial-like identity.

## Evolution of the auxiliary cardiac chambers

There exists a significant gap in our understanding of heart evolution with regards to the origins of the large auxiliary chamber-like vessels as well as the myocardial chambers, in large part due to the extreme rarity of fossilized heart specimens[96]. It has been hypothesized that an ancestral chordate heart resembles that of modern tunicates featuring a single myocardial chamber with bi-directional blood flow, due to bilateral pacemakers and BSs[1–3,68,69,97]. This blueprint was subsequently elaborated on to include two myocardial chambers and unidirectional flow in the earliest vertebrates[5,98,99]. Importantly, although there are histological analyses[1,2], and even transgenic reporter lines that mark cells within the BSs of tunicates[68], not much is otherwise known about the molecular nature of the BS. Our data show that both *Ciona* BSs express orthologs of markers found in SMCs of the zebrafish BA and SV, including *hox3* and *piezo*, as well as orthologs of EC markers *fli/erg* and *ets/pointed1*. However, compared to their *Ciona* homologs, which showed expression in both BSs, some zebrafish homologs, such as *six1a* and *etv2*, which are respectively expressed in the SMCs and ECs based on our scRNA-seq data, show asymmetric expression in either the BA or SV. Similarly, it is interesting that the *Ciona* cardiac tissues expressed pan-cardiac markers, such as *nk4*, the vertebrate *nkx2.5* ortholog[100], as well as orthologs of markers that are chamber-specific in vertebrates, including *irx1* and *nr2f*. One hypothesis resulting from the chamber-specific expression of these orthologs in vertebrates

compared to the *Ciona* hearts is that ancestral single chamber hearts had intermediate cardiac chamber identity and the specialization of these genes helped to drive or reinforce the emergence of the two chambers[3,101]. Despite comparative symmetry of gene expression in the BSs, we found some evidence of polarity within the *Ciona* hearts, as orthologous genes associated with the anterior SHF, in particular *tgfb2*, were expressed in the arterial pole of the PS[102–106]. However, we did not find orthologs of the venous pole/posterior SHF expressed in the SS, such as the *Ciona nr2f* ortholog[107,108] or Hedgehog signaling, suggesting this pole in tunicates lacks venous identity that subsequently appeared with the emergence of vertebrate multichambered hearts[13]. There has also been debate about whether or not tunicates have SMCs[109–111], with only recent reports indicating *Ciona* indeed have SM-like cells[112]. While our data suggest that the vertebrate orthologs of marker genes examined are expressed in SMCs, we have not been able to discern if the tunicate BSs express orthologs of canonical SMC genes based on the current annotation of the *Ciona* genome. However, some genes, like *elastins*, that are SMC markers did not appear until the vertebrate lineage[113], as well as true neural crest cells[97,114], which contribute to the smooth muscle in the BA and SV[55,56]. Therefore, additional analysis will be needed to assess the specific nature of cells in the BSs and if they are similar to SMCs in vertebrates. Our data indicate that without atria the *nr2f1a* mutant hearts overtly revert to an approximation of a single-chambered, bi-directional heart, as they effectively only have a single contractile chamber and functionally have bidirectional flow (regurgitation), which correlates with the adaptive thickening and partial arterialization of the SV. Thus, based on our combined comparative and *nr2f1a* mutant analysis, we posit that the BA and SV in vertebrates stemmed from BS-like structures in chordate ancestors, which showed greater equivalency to capacitate bidirectional shear stress, and that the BA and SV in early ancestral vertebrates simultaneously functionalized in response to the differences in shear stress sensed by these chambers due to the addition of a second myocardial chamber and unidirectional blood flow.

In evaluating the evolution of the auxiliary chambers, our analysis of the zebrafish and *Ciona* needs to be integrated with current hypotheses about the proposed evolution of the SV in vertebrates, in particular the transition in early-branching vertebrates. All vertebrates examined, except the majority of teleost fish, appear to have an SV that has a more significant contribution of CMs, which are proposed to be detectable via electrocardiograms (ECGs)[9]. This trait has led to the hypothesis that an SV containing CMs is the ancestral state of the vertebrate SV. Furthermore, an agnathan-like SV and multichambered heart has been considered to be at the root of vertebrate heart morphologies, in part because histological analysis of the agnathan SV and ECGs suggests that it contains CMs[87,88]. However, the extent of striated muscle within the agnathan SV is not well defined from the histological analysis[45,86,115], and the specific cellular composition of the SV in agnathans has not been verified with current molecular and transcriptomic analysis. Although it has been proposed that the CMs in the SV of agnathans are necessary for atrial filling[88,116,117], owing largely to the very limited activation and delay observed during ECGs of the SV in the hagfish species *Eptatretus cirrhatus*[9,118], more recent studies in agnathans have called into question the functional significance of these contractions due to the SV in cyclostomes lacking valves to prevent backflow into the systemic vasculature and an inability to replicate the ECG readings in other species of hagfish[86]. Furthermore, despite the proposed presence of limited striated muscle in the agnathan SV from histological analysis, anatomical data has led some to propose that the SV of agnathans may not really be homologous to the SV in later-branching vertebrates[45,86], but actually an analogous structure, as the positioning of the SV relative to the ventricle and atrium is not in sequence from inflow to outflow like what is found in all other vertebrate fish[84,96,119]. Instead, the SV in the agnathan hagfish is located between the ventricle and atrium connecting to their

uniquely elongated AVC[45,86,120]. The agnathan hearts and cardiovascular systems also have additional characteristics that suggest further divergence from most vertebrate fish hearts, including the presence of chromaffin cells[86,120–122], the absence of a BA in favor of an OFT consisting of just a conus arteriosus (CA) in lamprey, which is a muscular extension from the ventricular arterial pole present in sharks and teleosts[8,48,123–125], and additional contractile extra-cardiac regions called the caudal and portal hearts[88,126]. Therefore, contrasting analysis and characterization of the agnathan SV suggest it may not reflect some of the ancestral traits found in vertebrate hearts with respect to the auxiliary vessels or be the best model for the predicted transition from a tunicate-like, early-branching chordate heart.

While discrepancies in the characterization of the agnathan heart and its annumerated peculiarities have led some to intimate that agnathans are too unique to be used to draw evolutionary conclusions and attention is better focused on other clades of fish[86], the prevailing hypothesis is instead that the teleost SV is in fact uniquely derived[9]. This assertion relies primarily on the relative absence of myocardial contribution to the SV of most teleosts with respect to other vertebrates including the early-branching agnathans[115]. Inspection of the zebrafish SV-atrial border shows that like other vertebrates their SV does contain atrial CMs at the venous pole that are surrounded by SMCs and extend into the SV, although it could be argued it is minimal (Supplementary Fig. 21). This proposed reduction of atrialization in the SV is reminiscent to the reduced CA at the arterial pole in teleosts[8], supporting a hypothesis that cardiac muscle extending from the transition zones at both poles of the heart into the adjacent auxiliary vessels may have been greatly reduced in teleosts, if it is assumed that longer muscularized poles of the hearts are ancestral states[3,9,13]. Nevertheless, evidence suggests that although reduced, the OFT and IFT of the zebrafish both incorporate cardiac muscle similar to what is observed in other vertebrates. Integrating the histological data from *Ciona*[1,2,68] with our transcriptomic analysis supports a model whereby an ancestral chordate heart was flanked by BS-like structures that lacked myocardial integration. Hence, we currently propose two scenarios for the evolution of the SV and composition of the SV in early branching vertebrates: (1) greater CM incorporation into the SV reflects the ancestral state in vertebrates, with the teleost SV subsequently reverting to a state that reflects a hypothesized vertebrate ancestor that possessed more distinct myocardial and SV chambers similar to tunicates; (2) The agnathan SV is actually more derived and that the teleost heart reflects a symplesiomorphic trait with the tunicate-like hearts, with subsequent expansion of CM contribution and atrialization of the transition zone between the atrium and SV in later-branching vertebrates. Regardless of these evolutionary scenarios, we posit that the teleost heart, even if derived, reflects a more ancestral heart bauplan that transitioned from chordates where there is a greater distinction between the myocardial chambers and the auxiliary inflow and outflow chambers. Hence, teleosts, by virtue of exhibiting an SV largely devoid of CM, would serve as an appropriate link to the chordate ancestor that circulated its blood via a muscular pump flanked by two pressure sensing-vascular sinuses.

Future work will be aimed at capitalizing on the unique qualities of the adult *nr2f1a*^*aco* mutants to elucidate the specific molecular mechanisms underlying the vascular hypertensive response of the SV. Furthermore, we recognize there are limitations in drawing conclusions about chordate heart evolution from molecular comparisons of just tunicates and zebrafish, which leaves open other scenarios for the evolution of the inflow tract and cardiac auxiliary vessels. Thus, we will distinguish these evolutionary hypotheses of heart anatomy via significantly greater comparative analyses of additional urochordate, agnathans, and fish species focused on illuminating the specific cell types harbored within their auxiliary cardiac vessels.

## Methods

### Animal husbandry

All zebrafish husbandry was performed under standard laboratory conditions. WT zebrafish used were of the mixed AB/TL background. Transgenic lines used in this study include: *Tg(amhc:Cre-ERT2)*^*sd20*[127], *Tg(ubb:loxP-AmCyan-loxP-ZsYellow)*^*fb5* [128], *Tg(sox10:GAL4,UAS:Cre)*^*la232*[56], *Tg(-3.5ubb:loxP-EGFP-loxP-mCherry)*^*cz170*[156], *Tg(acta2:eGFP)*^*ca7*[129], and *Tg(kdrl:GFP)*^*s843*[130]. The allele designation for *nr2f1a*^*aco* is *nr2f1a*^*ci1017*, which was identified from an ENU screen. Genotyping was performed using primers (Supplementary Table 1) that produce a 172 bp product and create an EcoRI site in the mutant from a snp in the *nr2f1a* exon 2. Zebrafish were used at ages from 48 hpf through 1 year old (adults). Adult fish 2.0–2.5 cm were used for analysis in experiments. All experiments were carried out in accordance with guidelines established by Cincinnati Children's Hospital Medical Center Institutional Animal Care and Use Committee (IACUC) and the approved protocol IACUC2020-0091. Adult zebrafish were monitored daily by veterinary technicians at Cincinnati Children Hospital Medical Center (CCHMC).

Adult *Ciona robusta* (formerly called *Ciona intestinalis* Type A) between 5–7 months old with a size ranging from 5 to 10 cm were collected in the Gulf of Taranto, Italy, by hand picking at low depth. Animals were transported to the Stazione Zoologica Anton Dohrn in Napoli (SZN) with an animal transport van equipped with refrigerated tanks and supplied with an oxygen air pump. Once they arrived at the Stazione Zoologica, the animals were maintained in the animal facility with an open tank system of 150–200 Liters with these parameters: ~20 °C, pH ~7.8, salinity ~37.9 ‰ and with a photoperiod of 12 L:12D. Animals were fed every day with a solution of shellfish diet 1800™ Instant Algae®. Invertebrate organisms including *Ciona robusta*, a marine invertebrate, are not subject to institutional care committees in Italy and are not included in the law of DIRECTIVE 2010/63/EU OF THE EUROPEAN PARLIAMENT AND OF THE COUNCIL of 22 September 2010, which details the protection of animals used for scientific purposes. *Ciona robusta* as invertebrate animals are considered to not have the ability to experience discernible pain, suffering, distress, and lasting harm.

### Western blots of zebrafish embryos

Western blots were performed as previously reported[38]. Briefly, 25 embryos/condition were harvested at 24 hpf and homogenized in embryo lysis buffer (20 mM Tris - pH8, 50 mM NaCl, 2 mM EDTA, 1% NP-40) with cOmplete™ protease inhibitor (Roche). Two embryo equivalents were run on a 10% polyacrylamide gel. Western blotting was performed using standard methods. Zebrafish Nr2f1a antibody was used[38]. Monoclonal anti-α-tubulin (Tuba; Sigma) was used as a loading control. LI-COR IRDye® 680LT Donkey anti-rabbit secondary was used to visualize the anti-Nr2f1a antibody. LI-COR IRDye® 800CW Donkey anti-mouse secondary was used to visualize the anti-α-tubulin loading. Blots were imaged using an Odyssey CLx LI-COR imager. Additional antibody information is included in Supplementary Table 2.

### Quantification of embryonic cardiomyocytes

IHC of embryonic zebrafish hearts for quantification of cardiomyocytes was performed as previously described[38]. Primary antibodies used were: Amhc (Atrial myosin heavy chain; University of Iowa Developmental Studies Hybridoma Bank) and Living Colors® polyclonal anti-DsRed antibody (Clontech). Secondary antibodies used were anti-mouse IgG1-FITC (Southern Biotech) and anti-rabbit IgG-TRITC (Southern Biotech). A Zeiss M2BioV_12 Stereo microscope was used to image hearts. Additional antibody information is included in Supplementary Table 2.

### Harvesting of zebrafish hearts

The hearts from adult zebrafish were dissected similar to previous descriptions[131,132]. Adult fish used were all between 2.5–3.0 cm in length

and 6 months to 1.5 years post fertilization. Adult fish were euthanized in a beaker containing 0.16 µg/ml tricaine dissolved in system water until there was no gill movement. Fish were then placed in a damp, slotted sponge with their ventral side up. Iridectomy scissors were used to make a lateral incision perpendicular to the midline immediately anterior to the abdomen. An anterior incision was made along the midline to open the thoracic cavity and forceps were used to pinch the BA away from the ventral aorta. Finally, a slight anterior tension was applied to the BA to visualize the SV, and a second pair of forceps was used to sever the veins leading into the SV, thus separating the heart from the rest of the body. Isolated hearts were washed several times in 1X PBS before processing.

Juvenile fish (3–4 weeks post fertilization and ranging from 0.6–0.8 cm in length) were euthanized in a petri dish with tricaine in system water as above. Following the cessation of movement, forceps were used to expose the heart within the chest. The hearts were then removed by first pinching the distal connections of the SV and then pulling the BA away from the body of the fish. Isolated hearts were washed several times in 1X PBS before processing.

Hearts were isolated from larval fish (1–2 weeks post fertilization) using a method similar to previous descriptions[133]. Since the fish are still largely transparent, their hearts are visible. Euthanized larval fish in a petri dish (as above) were held with one pair of forceps. Another pair of forceps was then used to gently grasp the entire heart and pull it away from the body of the fish. The hearts were then separated from the pericardial tissue and fixed for analysis.

### Preparation of tissues for histology
Washed adult hearts were either flash-frozen with 2-methyl-butane submerged in liquid nitrogen or placed in 4% PFA in 1X PBS for fixation overnight at 4 °C. For histology of whole adult and juvenile fish, following euthanasia the hearts were exposed within the chest cavity, as described above, and the whole fish were fixed in 4% PFA in 1X PBS for fixation overnight at 4 °C. To prepare hearts and whole fish for cryosectioning, the samples were then briefly washed in 1X PBS before being subjected to a perfusion series of 15% sucrose/1X PBS and 30% sucrose/1X PBS. Hearts and whole fish were placed in the sucrose/PBS until they sank to the bottom of the container or overnight. The hearts and whole fish were then embedded in Optical Cutting Temperature (OCT) (VWR, 25608-930) compound blocks using disposable molds (Ted Pella, 27147-2) submerged in a supercooled slurry of ethanol and crushed dry ice. Sectioning was conducted at −20 °C using a Leica model CM 1860 cryostat. Adult fish and isolated hearts were sectioned at a thickness of 10 µm. Whole juvenile fish were sectioned at a thickness of 5 µm. Three slides were used per block alternating slides in every section to accomplish the most representative range of tissue on every slide. Three sagittal sections at the midline of whole embryos were used for analysis of the SV (Supplementary Fig. 22).

### Acid fuchsin orange G (AFOG) staining
AFOG staining was conducted on 10 µm cryosections similar to previous descriptions[134,135]. Sections were post-fixed in pre-heated Bouin's solution for 2 h at 60 °C and 1 h at room temperature (RT). The slides were then rinsed, mordanted by soaking in 1% phosphomolybdic acid, and then treated with trichrome AFOG (pH 1.09) solution containing 1× aniline blue, 2× orange G, and 3× acid fuchsin in dH$_2$O for 15 min. Finally, the sections were dehydrated in an ethanol series, cleared with Citrosolve, and mounted using 100 µl Cytoseal for imaging.

### Wholemount in situ hybridization
Wholemount in situ hybridization was performed on adult hearts and embryos using a standard protocol[136]. In brief, adult hearts and embryos were fixed overnight at 4 °C, dehydrated/rehydrated in a methanol-1X PBS/0.1% Tween-20 (PBST) series, and pre-hybridized in hybridization buffer (50% formamide, 5x saline-sodium citrate (SSC),

100 ug/mL heparin, 50 mg/mL torula RNA, 0.1% Tween 20, 9.2 mM citric acid) at 70 °C. Hearts and embryos were then incubated with anti-sense RNA probe for 2 days at 70 °C followed by a series of stringent washes at 70 °C. They were first washed in 50% formamide/2X SSC/0.1% Tween-20, followed next by a second wash in 2X SSC/0.1% Tween-20, and finished in 0.2X SSC/0.1% Tween 20. The hearts and embryos were then washed at RT with 1X PBS, followed by blocking with 10% sheep serum/1X PBS/0.1% Tween-20 and detection of the probe with an anti-Digoxigenin antibody (Roche, 11093274910) diluted 1:5,000. The color reaction contained equal parts 4-Nitro blue tetrazolium chloride (NBT) and 4-toluidine salt (BCIP) in a staining solution (0.1 M Tris-HCl pH 9.5, 0.05 M MgCl$_2$, 0.1 M NaCl, 0.1% Tween 20) (3.5 ml/ml each). Following the coloration reaction, hearts, and embryos were dehydrated and rehydrated with a methanol/1× PBST series and perfused in an increasing glycerol/1X PBST series for storage and imaging. Probes used were to *amhc* (ZDB-GENE-031112-1), *vmhc* (ZDB-GENE-991123-5), *nr2f1a* (ZDB-GENE-980526-115), *kdrl* (ZDB-GENE-000705-1), *notch1b* (ZDB-GENE-990415-183), and *flt4* (ZDB-GENE-980526-326).

### In situ hybridization on sectioned tissue
Slides to be stained were post-fixed in 4% PFA for 10 min and washed twice for 5 min in 1X PBS. Slides were treated for 90 s with a Proteinase K solution (17.5µl of 10 mg/mL ProK (Sigma, P2308) in 80 mL 1X PBS) and washed once in 1X PBS for 5 min before re-fixing the slides in 4% PFA for another 5 min. Next, slides were washed 3 times in 1X PBS for 5 min each time and dehydrated by submerging first in 70% ethanol for 5 min and then 95% ethanol for 2 min. Humidity chambers were constructed by lining 740 mL Tupperware containers with damp wet paper towels and placing plastic pipette tip box inserts on the paper towels to hold the slides. Ethanol was removed and slides were allowed to air dry in the jar for 5 min before being placed into humidity chambers for hybridization. RNA probes were diluted in a hybridization solution as above and heated to 80 °C before applying 300 µl of probe dilution to each slide. Parafilm was cut into strips large enough to cover the slides and gently placed on top of the probe solution to act as a coverslip to evenly treat each tissue section. Slides with the probe were incubated overnight at 65 °C in a laboratory oven (Quincy Labs, Model 10 Gravity Convection Analog Lab Oven). The following morning, coverslips were removed and slides were placed in a coplin jar containing a high stringency wash solution (100 mL water/125 mL formamide/25 mL 20X SSC) and incubated for 30 min in a 65 °C water bath. The slides were then washed in an RNAse buffer solution (50 mL 5 M NaCl/5 mL 1 M Tris-HCl pH7.5/ 5 mL 0.5 M EDTA pH8 in 440 mL water) in a 37 °C water bath 3 times for 10 min each wash. Slides were placed in an RNAse A solution (RNAse A (Qiagen, 19101) 1:1000 in RNAse buffer) for 30 min at 65 °C in a water bath followed by a final 15-min wash in RNAse buffer at 37 °C. The slides were then taken through a series of washes: high stringency solution (described above) 2 times for 20 min each at 65 °C, 2X SSC 1 time for 15 min at 37 °C, 0.1X SSC for 15 min at 37 °C, and PBST for at least 15 min at RT. After all of the washes, the slides were blocked as above in humidity chambers for 2 h, washed 4 times in PBST for 5 min each, and treated with 300 µl of anti-digoxigenin antibody diluted 1:2500 in blocking solution in humid chambers at 4 °C overnight. Antibody solution was washed off the slides via 6 washes in PBST for 1 h each wash. Finally, slides were treated in humidity chambers with staining solution as above (300 µl each) 2X for 10 min, treated with NBT/BCIP in staining solution as above to develop the color reaction, and washed in 1X PBS before mounting with a glass coverslip. Slides were imaged with a Zeiss Imager.A2 compound microscope.

### Wholemount IHC
Hearts isolated from juvenile fish were subjected to wholemount immunostaining as reported previously[133]. In short, square wells were drawn on a coverslip using a hydrophobic PAP pen into which

4% PFA/1X PBS fixative solution was pipetted. Individual hearts were placed in these wells and fixed for 30 min, washed in 1X PBS supplemented with Tween-20, and blocked for 1h with 10% sheep serum solution. Primary antibodies were applied for 1h at RT in blocking solution and washed 3 times in PBST for 5 min each before treatment with secondary antibodies in blocking solution for 30 min at RT. Following a final wash in 1X PBST, hearts were imaged on the same coverslip free-floating in PBST using a Nikon A1R inverted confocal microscope.

Immunostaining was performed on sectioned tissue following a stringent heat-induced antigen retrieval step utilizing citric acid (1% in research grade water) for 5 min in a pressure cooker set to high pressure and heat. Sections were allowed to cool to RT, permeabilized using 0.2% Triton X-100 in 1X PBS, and blocked with 1% BSA/1% heat-inactivated goat serum/0.025% Tween-20/1XPBS. Primary antibodies were applied to the slides in the blocking solution overnight at 4 °C. Slides were washed for 5 min in 1X PBS before applying secondary antibodies in the blocking solution were applied to the slides for 30 min at RT. Slides were then mounted in 100 μl ProLong anti-fade medium (Invitrogen, cat # P36930). Antibodies used were: MF20, S46, Vmhc, Elastin, Mylk, and anti-RCFP. See Supplementary Table 2 for additional antibody information. Slides were imaged using a Nikon A1R inverted confocal microscope.

### Lineage tracing and tamoxifen treatment

Heterozygous $nr2f1a^{aco}$ fish hemizygous for either $Tg(amhc:Cre^{ERT2})$ or $Tg(ubb:CSY)$ transgenes were crossed. The resulting embryos were taken at the shield stage (6 hpf) and placed in glass vials with 1 mL embryo water (maximum 30 embryos per vial). A 50 mM 4-hydroxytamoxifen (4-HT) dissolved in ethanol was diluted to a working concentration of 10 μM in glass vials with embryo water. Vials were then placed on a nutator at 28.5 °C with loosened lids until 48 hpf. The WT sibling and $nr2f1a^{aco}$ mutant embryos were then sorted based on phenotype. WT and $nr2f1a^{aco}$ mutant embryos were raised until they were adults, the hearts harvested, prepared for histology, and immunostaining, as described above.

### Intrathoracic EdU labeling

Intrathoracic EdU labeling of the SV was performed similar to previous reports when it was used to label CMs[137]. Holding chambers with trenches were made with 2% agarose (w/v) in embryo water. Juvenile fish were anesthetized using tricaine diluted in system water to a concentration of 0.16 μg/mL until gills almost stop moving and were placed in the chamber, which was filled with embryo water and tricaine at the same concentration, on their back for the duration of the injection process. A fine glass capillary needle was used to pierce the thoracic cavity and inject 2 nL EdU solution (1 mM in DMSO) at 21 and 24 dpf. The hearts from the EdU-injected fish were harvested at 28 dpf. EdU was detected using the Click-iT EdU Cell Proliferation Kit (Thermo Fisher, C10337) on cryosections coupled with immunohistochemistry as described above.

### Echocardiography

Echocardiography was performed similar to previous reports except with the following modifications[58,138]. Adult fish were anesthetized using 0.16 μg/mL tricaine and placed on their back in a clay holder submerged in a container with system water. A high-frequency transducer (Visual Sonics, MS700D, 40–50 MHz) was lowered directly into the water at a distance above the chest of the fish that empirically produced an optimal image. The Vevo2100 Imaging System was used to capture echo videos and two-color doppler functionality was utilized to pseudo-color the direction of blood flow.

### Drug treatments

For Captopril treatments, WT and $nr2f1a^{aco}$ mutant larvae were placed in system water with 250 μM Captopril (diluted from 250 mM stock in DMSO; Sigma-Adlrich, C4042) or vehicle (DMSO) at 14 dpf. For Enalapril treatments, embryo water containing 50 μM Enalapril (diluted from 100 mM stock in ethanol; Sigma-Aldrich, E6888) or vehicle (ethanol) was applied to WT and $nr2f1a^{aco}$ mutant embryos in a petri dish beginning at 48 hpf. For estradiol treatments, embryo water containing 10 nM β-Estradiol (diluted from 50 mM stock in DMSO; Sigma-Aldrich, E8875) or vehicle (DMSO) was applied to WT and $nr2f1a^{aco}$ mutant embryos in a petri dish beginning at 48 hpf. WT control larvae and embryos were treated with a corresponding relative volume of the vehicle. For Enalapril and estradiol, embryos were transitioned to system water at 6 dpf. Water containing the drugs or vehicle controls was refreshed every 48 h. Treatments for all drugs were continuous until 28 dpf, when fish were harvested and processed for histology and immunostaining. Fish were measured from snout to base of tail to ensure they were of equal size for analysis. Cryosectioned tissue was stained for Elnb, and imaged with a Nikon A1R inverted confocal microscope, and ImageJ was used to measure the thickness of the sinus walls. 10 measurements were taken per section and 3 sections were analyzed per fish.

### Exercising of adult and juvenile zebrafish with the swim tunnel

Adult zebrafish were placed in a mini Loligo swim tunnel system (170 mL 110 V/60 Hz, Product number SY10600) and allowed to briefly acclimate to the chamber. The flow was then initiated at a low velocity (~4 cm/s or 150 rpm) for 10 min to allow the fish to warmup. For sprint challenges, following the warmup, the propeller speed was increased by 50 rpm (~2.5 cm/s) every 1 min until the fish could not swim against the current, i.e. the fish lay flat against the backstop for 5 s. The stamina challenges were conducted in the same manner except the propeller speed was increased every 10 min until failure.

For exercising juvenile $nr2f1a^{aco}$ mutant fish beginning at 21 dpf, a fine mesh was placed in front of the backstop due to their smaller size to prevent them from passing through the openings. The juvenile fish were placed in the swim chamber and allowed to briefly acclimate. They were then allowed to warm up at a low velocity (above) for 10 min prior to pacing. Fish were then exercised for 30 min at the highest speed they could manage without being pressed against the backstop. Exercise was repeated every day for 2 weeks at which time juvenile fish were processed for histological analysis.

### Preparation of tissues and analysis for quantitative PCR (qPCR) and bulk RNA sequencing

Hearts were harvested from adult fish as described above. Fine forceps were then used to gently separate the individual heart components (atrium, ventricle, BA, and SV) in 1X PBS before being pooled (20 hearts for each genotype) and placed in a TRIzol solution (Invitrogen, 15596026). Total RNA was extracted from these pooled samples using a standard phenol chloroform phase-separation method followed by passing the supernatant through a filter column using the stepwise protocol included in the PureLink RNA Micro Kit (Invitrogen, 12183016). Individual cDNA samples were generated using 1μg of total RNA isolated from each chamber by following the ThermoScript RT-PCR System kit (Invitrogen, 11146-016) protocol. qPCR was carried out in triplicate using targeted primers and Power SYBR Green PCR Master Mix (Applied Biosystems, 4368706) under standard PCR conditions in a Bio-Rad CFX PCR machine. Resultant expression levels were standardized to the ubiquitously expressed $actb1$ gene and the Livak Method[139] was used to calculate the average fold change in expression ($2^{-\Delta\Delta CT}$). Primer sequences are detailed in Supplementary Table 1.

Zebrafish RNA-seq reads were aligned to the UCSC zebrafish genome, danRer11, using STAR aligner(v2.7.4)[140] after trimming the Illumina adapter using cutadapt(v2.1)[141]. These libraries were prepped across multiple batches with different library types in terms of RNA strand and sequencing mode (single-end or paired-end). Therefore, we treated them as single-end libraries by choosing read1 or read2 to

match the strand as much as possible, and batch correction was applied as described below. Only uniquely aligned reads were retained for downstream analysis. Gene expression levels were quantified as raw read counts using FeatureCounts[142] in the subread(v1.6.2) package with an option, "-O --fracOverlap 0.8", and with a proper strand option (-s). Given the mixed sequencing batches and library types, we incorporated RUVseq(v1.28.0)[143] batch correction (RUVs, $k = 1$) in DESeq2(v1.34.0)[144] differential analysis. Differentially expressed genes were identified by FDR < 0.05. Hierarchical clustering was performed using the Pearson correlation coefficient as a similarity measure under Ward's criterion, and a heatmap was visualized in $z$-score. Data used for analysis are publicly available in GEO (Accession #: GSE195548).

### Preparation of cells and analysis of single cell RNA sequencing
Single cell preparations were prepared similar to previously published protocols designed to dissociate rigid elastic tissue with minor variations[43,145]. The SV and BA were dissected from the hearts of 30 WT and 24 *nr2f1a* mutants using fine forceps and pooled in a Tyrode's solution (pH 7.4) following washing in 1X PBS. The pooled tissues were washed 3 times in Tyrode's before the addition of Liberase IV (0.29U/ml, Roche, 05401020001), Elastase (1.0 U/ml, Sigma, E7885), and Pronase E (0.92 U/ml, Sigma, P5147). Dissociation of the tissues was facilitated via incubation at 37 °C and gentle trituration with pipettes of descending tip diameter (first 2.0 mm followed by 0.8 mm. Can also use filtered P1000 and P20 tips, respectively). The dissociation reaction was halted by transferring the cells to a modified Kraft-Brühe solution (pH 6.9) and the cells were spun down to a pellet before being resuspended in a final 1% BSA solution in 1X PBS. Single cells were submitted to the CCHMC Gene Expression Core for 10x Genomics analysis utilizing Chromium instrumentation. -13,000 cells were submitted for sequencing from each sample which yielded captures of 13,004 WT BA cells (32,642 reads/cell), 13,439 WT SV cells (27,137 reads/cell), 12,736 mutant SV cells (33,327 reads/cell), and 14,095 mutant BA cells (29,673 reads/cell).

The scRNA-Seq FASTQ files (10x Genomics 3′ version 3) were aligned to the Zebrafish Ensembl version 91 reference genome (GRCz11) and transcriptome, using Cell Ranger version 3.1.0. Cell Ranger filtered feature sparse matrix counts files from all samples (HDF5 format) were supplied to AltAnalyze[146] version 2.1.4 for unsupervised analysis using the EnsMart 91 zebrafish database and ICGS2 algorithm with default parameters. Cell-type predictions were initially obtained from ICGS2 based on marker enrichment and then further refined based on manual curation with literature-associated markers. The obtained ICGS2 clusters were filtered to include muscle, endothelial, and epithelial clusters for downstream UMAP visualization. Data used for analysis are publicly available in GEO (Accession #: GSE229821). To confirm the assignment of mutant cells to WT clusters, we performed supervised classification of all mutant capture cells to WT cells and clusters using the software cellHarmony[147]. Supervised embedding of these mutant cells into a WT restricted UMAP was performed using scikit-learn to train and transform the ICGS2 marker genes using the top 50 PCA components (UMAP-transform). All differential expression analyses were performed using the software cellHarmony in AltAnalyze (fold > 1.2 and empirical Bayes $t$-test $p < 0.05$, FDR corrected), with secondary gene-set enrichment analyses and visualization performed in GO-Elite.

For pseudotime analysis, we applied both Monocle 2 and SlingShot. Log-transformed gene-by-cell expression file and cell type labels file from all samples generated from ICGS2 were filtered for smooth muscle cells (Clusters 1-8) and were provided as the input to Monocle 2(v2.13.0)[148]. The log-transformed file was exponentiated and modeled with negative binomial distribution using Monocle 2 ('expressionFamily = negbinomial.size'). Monocle 2 was allowed to select its own genes for pseudotime estimation based on differential gene analysis across the filtered smooth muscle ICGS2 groups ('fullModelFormulaStr = ~Groups'). The reverse graph embedding (RGE) method ('method' in reduceDimension) was set to "DDRTree" as recommended by the authors of Monocle 2. For SlingShot(v1.8.0)[149], we embedded the gene expression data for the selected populations using the software SPRING[150], using programmatic defaults for both tools.

### Preparation and analysis of bulk RNA-sequencing of *Ciona robusta* hearts
Hearts were dissected from adult *Ciona robusta* in cold filtered sea water using surgical scissors under a Zeiss Stemi 2000/2000C Stereomicroscope and immediately placed in RNA*later* (Invitrogen, AM7020) at 4 C°. After 3 days in RNA*later*, the whole hearts were then placed in new RNA*later* solution and dissected into the 3 parts: the pharyngeal blood sinus (PS), which is connected to the subendostylar sinus and is also called the hypobranchial vessel; the blood sinus adjacent to the stomach (SS); the central tubular part of the heart (C). Pericardial cells were removed. The blood sinus tissues each had a small part of the adjacent tubular myocardium. The sections from 15 hearts were pooled. The pooled hearts were then immersed in 300 μl of lysis buffer from a RNeasy Micro kit (Qiagen, 74004) and sonicated with the Branson Digital Sonifier on ice with an amplitude of 20% for 20 s. Tissue sonication was repeated 3 times with intervals of at least 30 s. Following sonication, the samples were centrifuged for 10 min at the max speed at 4 °C. The supernatant was placed in a new tube and processed for the RNA extraction according to the RNeasy Micro kit instructions. The isolated RNA samples were checked for quality and integrity on 1% agarose gel and Bioanalyzer. 1 μg of each RNA has been then processed for the RNA-seq analyses. Samples were submitted to Genomix4 Life (Salerno, Italy) for sequencing. *Ciona* RNA-seq data was analyzed in the same way as indicated above for zebrafish RNA-seq analysis except using the UCSC *Ciona* genome ci3. Differential genes were identified by *P*-value < 0.05 for hierarchical clustering. Data used for analysis are publicly available in GEO (Accession #: GSE195549). Orthology for genes was determined using DAVID Bioinformatic Resources (david.ncifcrf.gov) and the BioMart tools in Ensembl (ensemble.org).

### Statistics and reproducibility
The experiments described herein were not randomized and the researchers were not blinded during data analysis. Sample sizes for experiments were not predetermined using power analysis. Prism 9 software was used to perform statistical comparisons other than for RNA-seq analysis. For comparisons between two conditions unpaired, two-sided Student's $t$-tests were used. For comparisons between multiple conditions, an ordinary one-way ANOVA with multiple comparisons was used. A $P$ value of <0.05 was interpreted as being statistically significant. Results are represented as the mean ± SEM. For representative images of adult hearts and embryos in Fig. 1a and Supplementary Fig. 1c, >100 WT and *nr2f1a*[aco] mutants have been examined over >3 experiments. For representative images of histological analysis in Figs. 1c–e, 1j–n, 2a, and Supplementary Figs. 1c, 2a, 2b, 3, sections from at least 3 WT and *nr2f1a*[aco] mutants in 3 independent experiments.

### Reporting summary
Further information on research design is available in the Nature Portfolio Reporting Summary linked to this article.

## Data availability
All data generated and used for analysis in this study are available in the manuscript or Source Data files. Source Data are provided with this paper. Sequencing data has been deposited in GEO Accession #'s GSE195548, GSE195549, and GSE229821. There are not restrictions on availability of the data. Source data are provided with this paper.

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

## Acknowledgements

This manuscript was funded by the National Institutes of Health grants R01 HL137766, R01 HL141186, and R01 HL154522 to J.S.W., and Ruth L. Kirschstein Predoctoral Individual National Research Service Award F31 HL147399 and Training Grant T32 HL125204 to J.T.G.

## Author contributions

J.T.G. performed formal analysis of all experiments examining the hearts in juvenile and adult nr2f1a mutants and garnered tissue, cells, and RNA for Next Gen bulk and single cell sequencing experiments. E.D. collected Ciona tissues samples and RNA. P.R. sequenced the deletion in the nr2f1a mutant allele. J.T.G. and R.J.V. performed echocardiography on zebrafish. H.W.L. performed bioinformatic analysis of the bulk RNA samples from zebrafish and Ciona. K.T. performed bioinformatic analysis of the single cell RNA samples. N.S. performed and supervised single cell RNA sequencing analysis. J.TG. and J.S.W conceptualized the project and interpreted the experiments. J.T.G. generated the figures. J.S.W. supervised experiments. J.T.G. and J.S.W. wrote the majority of the manuscript except for parts of the methods, which were provided by H.W.L and N.S. All authors commented on the manuscript and agreed to its final submitted version.

## Competing interests

The authors declares no competing interests.
