## [Peer Review File · Nature Communications]

Sinus venosus adaptation models prolonged cardiovascular disease and reveals insights into evolutionary transitions of the vertebrate heartReviewers' comments:

Reviewer #1 (Remarks to the Author):

Summary

In this manuscript, Gafranek et al. report on their phenotypic characterization of acorn worm, a novel mutation in the transcription factor coding gene *Nr2f1a* in zebrafish. The mutation is presumably hypomorphic, allowing homozygous animals to survive to adulthood. The authors convincingly demonstrate that the atrium is altogether lacking, while the hearts of *Nr2f1a*aco mutants display enlarged sinus venosus (SV). Through elegant genetic lineage tracing and clever physiological treatments and assays, the authors demonstrate that this enlargement does not result from integration of atrial cardiomyocytes into the SV, but instead develops as a result of aberrant hemodynamics. Through bulk and single cell RNA-seq, the authors characterize the cellular composition and gene expression of the main cardiac compartments (bulbus arteriosus, BA; ventricle, V; atrium, A; and sinus venosus, SV) in both wild-type controls and mutant animals. Despite certain short-comings in the analyses (see below), these profiling assays do not reveal substantial changes in the cellular compositions of BA, V and SV compartments, but instead a relative "arterialization" of the SV in *Nr2f1a*aco mutants, whereby SV transcriptomes resemble their BA counterparts more so than in controls. This intriguing observation led the authors to use bulk RNA-seq of dissected tissues and ask if the presumably equivalent blood sinuses flanking the tunicate heart also share a greater similarity with each other than with the central myocardium.

Key merits

This focused manuscript has several key merits that warrant publication, including the characterization of the *Nr2f1a*aco phenotype, from developmental, physiological and transcriptomic perspectives, and the intriguing observations that BA and SV share under-appreciated gene expression similarities, especially in the *Nr2f1a*aco mutant, as also observed in tunicates, which hints at a possible deep origin of shared features of cardiac blood sinuses.

The study also provides rich transcriptomic datasets for future work.

General evaluation

In summary, this is an original and meritorious paper, with a suite of solid data and an intriguing hypothesis concerning the deep evolutionary origins of cardiac blood sinus.

Nonetheless, several concerns remain, which warrant attention to further improve the manuscript:

Concerns

- The characterization of hypomorphic mutations could be improved, especially compared to null mutations.
- It is not entirely clear what happens to cells that would normally form the atrium in the *Nr2f1a*aco mutant. Are they integrated into the ventricle (this seems to be the case), not proliferating and/or dying altogether?
- In Fig. 4, WT and mutant samples on same UMAP seem markedly disjointed. It is worth noting in the main text if techniques such as anchor-based alignment were performed to integrate the data from the two genotypes. The methods section makes no mention of this. In the absence of "alignment", the differential gene expression analysis may be affected. This also casts some doubts about statements in LL.170-172.
- The conclusion (LL. 175-177) should be amended as it is not clear whether the observed differences in gene expression can be attributed to the mutation, its developmental consequences and/or the differences in hemodynamics. More extensive studies with experimental design disentangling the primary developmental defects from secondary, hemodynamically-driven, defects would be needed to solve this question.
- The convergence/arterialization of SV in mutants should be substantiated by more statistically robust analyses (e.g. mutual enrichment using the detected gene set as background), in addition to

the few shared markers highlighted.

- Likewise, the similarity between blood sinuses in *Ciona* is circumstantial in the absence of statistical analysis showing significant mutual enrichment of regional markers, and their conservation in zebrafish.
 - The orthology assignment between *Ciona* and zebrafish genes are not explained. This is important to conclude about conserved regulatory programs for
 - Discussion of smooth muscle vs endothelial cells, and potential pacemaker cells at either side of the *Ciona* heart tube are worth discussing..
-
- fig.1f shows similarity only for a few genes. A PCA analysis (as shown in Ext Data Fig. 7, which could be moved to the main figure) PC plots of samples would be more convincing and/or biclustering analysis of conditions and genes
 - Fig.1j. Indicate marker used for neural crest, not NC:mCherry

Reviewer #2 (Remarks to the Author):

Adaptation of the sinus venosus reveals insights into evolutionary transitions of the vertebrate heart by Gafranek et al.

In this study, nr2f1a hypomorphic mutants are being investigated. Like nr2f1a mutants, these fish show atrial dysmorphogenesis, but a fraction of the fish survive and have been studied here. The fish show absence of atrial muscle, and an enlarged sinus venosus seems to be connected to the ventricle. Transcriptomes of mutant sinus venosus "replacing" the atrial compartment resembles both the sinus venosus of wild type fish and, to some extent, the outflow tract (bulbus arteriosus). The mutant sinus venosus is subject to retrograde blood flow, and altering sheer stress impacted on wall thickness. Single cell RNA sequencing suggested that mutant sinus venosus smooth muscle cells had differentiation state similar to those of both wt and mutant bulbus arteriosus. They also compared the transcriptomes with those of *Ciona*, and found indications supporting the ancestral equivalency between the large vessels that flank the vertebrate heart. The authors claim their "data reveal that the foundation of a rudimentary chordate heart with equivalent flanking vessel-like chambers remains latent within basal vertebrates, and hint at evolutionary origins underlying vascular disease".

Comments

1) An important assumption underlying this paper is that the sinus venosus of the zebrafish heart represents that of vertebrates, and therefore can be used as model to gain insight into evolutionary origins. However, the sinus venosus of zebrafish is vascular, whereas the sinus venosus of most other vertebrate species investigated is cardio-muscular. Therefore, the zebrafish sinus venosus can be considered to be specialized, in that respect not representing the sinus venosus most commonly encountered in vertebrate species. This derived state of the sinus venosus makes any comparison with "ancestral states" found in *Ciona* difficult at best. In fact, the authors themselves raise this point in line 81, but do not take into account the implication for their analysis and conclusions.

2) To me it is not clear how the atrial muscle has disappeared in the mature mutant fish. Is it not formed, the sinus venosus then by definition connected to the ventricle? Figure 1a suggests an atrial compartment-like structure is formed, but devoid of cardiomyocytes. Then, did the atrial cardiomyocytes disappear? In that case, the cardiomyocytes may have "simply" be replaced by fibroblasts and myofibroblasts and smooth muscle, not unlike what is seen in atrial fibrillation and aging. The "atrial" compartment may now adopt a vascular phenotype, and called "sinus venosus" by the authors. Or is it enlargement of the sinus venosus compensating for the (gradual) disappearance of the atrial compartment? Please clarify.

3) The authors seem to push the idea that the sinus venosus of the mutant adopts a phenotype resembling that of the bulbus arteriosus. Most comparisons in this study take into account a few specific transcripts that would show resemblance between mutant SV and the BA (e.g. fig 1f, etc). The selection of genes does not represent the actual transcriptomes, and should be replaced by a

hypothesis free unbiased analysis of representative transcriptomes. Indeed, the PC analysis of the RNAseq data and the sc-RNAseq data indicates that the mutant sinus venosus mostly resembles wt sinus venosus, with only selected markers indicating resemblance with bulbus arteriosus. In fact, figure 4c indicates a larger difference between wt and mutant bulbus arteriosus.

4) Line 105 (p5). This is incorrect. Atrial cells do not invade the vascular venous pole. The atria and subsequently the myocardial sinus venosus are formed from caudal second heart field progenitors in chronological order. Mouse lineage studies have shown that once differentiated to atrial muscle, these cells do not invade sinus venosus, and sinus venosus myocardium does not invade the venous vasculature.

5) Lines 175-177. This seems to be an overinterpretation. Did the authors study molecular identity in response to aberrant hemodynamic forces? These data are not provided as far as I can see.

6) An issue that the authors may want to address: It is possible that the changed flow impacts on the vascular sinus venosus of mutants, thereby imposing some outflow (arterial) vessel-like phenotypic aspects. This is an indirect consequence of AV valve dysfunction (I assume). Shear stress effects on vascular (arterial/venous) tissue have been well documented. But how does this finding have implications for atrial/sinus venosus evolution?

7) The value and implications of the comparison with *Ciona* are not clear to me. Selected genes point to resemblance of vascular poles of *Ciona* and zebrafish. Indeed, both species have vascular poles, showing resemblance. How is that novel, and how does that provide insight into evolutionary relationships? (see also point 1)

8) With the loss of the atrial compartment, the AV canal myocardium (positive for a and v myosin) seems to be expanded in mutants. Please comment on this.

9) An alternative explanation for authors findings is that *Nrf2f1a* imposes venous identity by directly regulating Notch signaling. In Nature 2005, You et al (PMID: 15875024) observed in mice: "Ablation of COUP-TFII in endothelial cells enables veins to acquire arterial characteristics". Something very similar may be happening in the mutant fish sinus venosus. This provides a good explanation for many of the observations by the authors. The link between venous pole (including atrial) identity and evolution and *Nr2f/Coup-TF* has been implicated already in literature. It remains unclear to me how authors findings provide more insight into venous pole evolution, and the implications for vascular disease seem unsupported by the provided data.

Reviewer #3 (Remarks to the Author):

General comments

The study by Grafanek et al reports on the phenotype in zebrafish of loss of *nr2f1a* (*Coup-tfII* in mammals) translation, with approximately half of the fish living to adulthood. In the dying fish, there are several signs of cardiovascular disease, but we are not told whether the loss of *nr2f1a* has effects outside of the heart. In mouse, loss of *Coup-tfII* has pronounced effects on the extracardiac vasculature. While there is a clear cardiac phenotype, such as absence of myocardium in the atrium, we do not gain much insight to what extent the vasculature is dysfunctional and the cause and effect of the pathology is quite obscure. Atrioventricular valve insufficiency is clearly documented. An attempt to link this to altered shear stress is made by estradiol treatment which associates with vasodilation. The experiment, however, is at best very indirect since this steroid hormone has profound transcriptional effects in a broad range of tissues. Bulk and single cell RNA sequencing is used to compare different compartments of the heart of WT and *nr2f1a* mutant fish, but it is a major short-coming to these experiments that no systemic vessels are included as benchmark reference. This in turn obscures the stated aim of the study which is to gain insight into how the chambered heart evolved. It may indeed be possible to unmask ancestral gene programs in zebrafish (Hawkins et al 2021 Cell). The present manuscript, however, does not have a gold-standard reference to what could be unmasked and a dysfunctional heart does not make a compelling case. Many of the reported

experiments have short-comings in design and, or, analysis – please see specific points below – and so while in aggregate an appreciable amount of work has been done, the manuscript at present does not have a core of solid findings.

Major points

1. This study on the evolution of the vertebrate heart has a well-argued premise, namely that there is a dearth in hearts that could represent transitional stages between the looped heart tube of sea squirts and the chambered hearts of vertebrates. It is a simplification, however, to reduce the heart chambers of agnathans to comprise an atrium and a ventricle only, since the sinus venosus of these hearts also contain myocardium and have distinct deflections on the ECG. Confusingly, you acknowledge this on L80-81 and correctly you are citing your reference 23 to support this. The evolutionary origins of the bulbus arteriosus is debated, namely whether it evolved in cartilaginous fishes or in teleost fish (e.g. Moriyama et al versus Lorenzale et al), but I do not know of any study that argues that the agnathan outflow tract is homologous to the bulbus arteriosus. My major concern is then that the study has failed to realise that the teleost heart is highly specialized by the absence of, or at least extreme reduction in myocardium, in the sinus venosus and the outflow tract.

2. Your use of the term 'chamber' is without definition and it is clearly at odds with a very large fraction of the literature. For example, a chamber should have myocardium and have electrical activity that is distinct in time from that of the other chambers. Defined such, the sinus venosus of all fish have a sinus venosus chamber, except some teleosts! On the other hand, the bulbus arteriosus, when defined as the bulb-shaped structure that gives rise to the name of it, does not have myocardium and it is without the distinct electrical activation and repolarization seen in chambers, so it is not a cardiac chamber but an arterial structure.

3. L78-80. I fail to see the logic here. Why would the loss of atrial identity / chamber formation, promote sinus venosus formation? It is perhaps an interesting hypothesis that the components upstream of the ventricle were vascular in a hypothetical ventricular/one-chambered heart evolutionary stage and that the atrial identity has been imposed on top of that. If so, you could articulate this, but other hypotheses could also explain the appearance of more than one chamber.

4. Your bulk RNAseq is indicative of similarity between the vascular/non-chamber sinus venosus and the bulbus arteriosus. This similarity may seem striking in comparisons to the very different tissues of actual chambers (atrium and ventricle), but in fact proper reference tissue have not been included, say a large artery and a large vein. Was the mutant bulbus arteriosus not included (Figure 1f) and if so, why not? Fig 1f: Why were only these 10 genes specifically selected for analysis? How significant is their differential expression in mutant tissues? Overview of what this mutation does to the atrial/SV/BA/etc transcriptome, GO-term analysis?

5. L104-110. I don't understand on what grounds/published data that you would hypothesize that atrial tissue contributes to the SV - and your data also bears out that this hypothesis is ill-founded. Also, given point 4 above, it is not clear to me that this part informs us on the role of the non-myocardial sinus venosus in contributing to the aberrant atrial segment.

6. Figure 2a, how come the bulbus cannot be seen on all images? Figure 2c-d, the indicated widths appear exaggerated to the actual wall thickness, and please use absolute units for the length measurements, say μm . At present, this analysis is short of convincing, and so are the similar analyses of Figure 3. Figure 2j, it appears to show that the Nr2f1a mutants have myocardium in the atrium, in contrast to what is stated on the basis of Figure 1a – please explain. In addition, the images of Fig 2c-j 3b lack anatomical context, it is unclear whether the authors are comparing similar regions. Fig 2d, 3c, and 3f: In these figures similar comparisons are made regarding the thickness of the SV wall. Why did the experiment shown in 3f call for significantly more samples than 2d and 3c? Why the experiments regarding remodeling performed at different developmental stages?

7. Estradiol, a steroid hormone, must be expected to affect transcription in numerous tissues and, concerning shear stress, its effects are not likely to be specific. So while the treatment reveals a difference in response between the WT and mutant, I don't see how these experiments inform us on the role of shear stress and blood flow more generally in causing the phenotype.

8. Your single cell analysis seems to suggest that the transcriptome of the bulbus is more affected than that of the non-myocardial sinus venosus. We must assume that Nr2f1a is not expressed in arterial vessels, but have you validated this and if so how do you explain the pronounced effect of loss of Nr2f1a on the identity of the bulbus? Is it not the bulbus that becomes more sinus venosus like than the other way around? Again, having extrapericardial vasculature as reference point appears to be important. For the time being, your experiments are not set up to differentiate between the possibilities that the bulbus and sinus are convergent on a shared vascular phenotype versus that one, or both tissues, are becoming more like each other. In your principal component analysis, please clarify how much of the variance is explained by the two components.

Minor comments.

L58: Can you articulate the reasoning for the mutant name 'acorn worm'? Given you are trying to understand early vertebrate evolution it is a bit too close to confusing, I would suggest, to name your mutant after hemichordates – are you implying the mutant heart resembles a heart-like structure in acorn worms?

L114: "was overtly comparable" – I don't think I understand what this means.

You make reference to, but do not provide, Fig 1m, Fig 5i.

Response to Reviewer Comments – Manuscript NCOMMS-22-16391A-Z:

We appreciate the insightful critiques of our original manuscript from all 3 reviewers. Reviewer 1's statements included that "*this is an original and meritorious paper, with a suite of solid data and an intriguing hypothesis concerning the deep evolutionary origins of cardiac blood sinus.*" Reviewer 3's statements included that the "*study on the evolution of the vertebrate heart has a well-argued premise.*" While the reviewers brought up positive attributes of the original manuscript, they all mentioned concerns that needed to be addressed. In particular, Reviewers 2 and 3 seemed to primarily have concerns about interpretations of our data with regard to the evolutionary hypothesis of the heart and sinus venosus (SV). Reviewer 3 also stated concerns about potential vascular defects in the *nr2f1a* mutants and comparisons of the bulbus arteriosus (BA) and SV. The revised manuscript has been substantially changed in response to the Reviewers' comments. Furthermore, we have directly addressed all of the Reviewers' specific comments with these changes in the revised manuscript. In considering the Reviewers' comments, we think the shorter format that the original manuscript was written in did not help the presentation of our data and allow for proper discussion and consideration of many of the ideas that were clearly necessitated. Therefore, we have significantly expanded the text with respect to rationale for experiments and the discussion of many of the specific points with regard to both the modeling of cardiovascular disease using the mutant and how our data integrate with different evolutionary hypotheses. In addition, we have modified the title to reflect both these points of our study. Undoubtedly, the changes made in response to the Reviewers' critiques of the original manuscript have allowed us to significantly clarify and improve the major points and messages of the manuscript. The additional data and analysis we have performed have helped clarify our points and further support our hypotheses. We have provided a point-by-point response to each of the 3 Reviewers' comments. Responses are indicated below in blue following the Reviewers' original specific comments. Less lengthy changes made to the text in the revised manuscript in direct response to the Reviewers' comments are indicated in gray. However, because of the extensive changes and significant addition of text to some parts of the manuscript, such as the Discussion, these were not highlighted as they did not provide clarity regarding changes made in response to the Reviewers' comments.

Reviewers' comments:

Reviewer #1 (Remarks to the Author):

Summary

In this manuscript, Gafranek et al. report on their phenotypic characterization of acorn worm, a novel mutation in the transcription factor coding gene *Nr2f1a* in zebrafish. The mutation is presumably hypomorphic, allowing homozygous animals to survive to adulthood. The authors convincingly demonstrate that the atrium is altogether lacking, while the hearts of *Nr2f1a*aco mutants display enlarged sinus venosus (SV). Through elegant genetic lineage tracing and clever physiological treatments and assays, the authors demonstrate that this enlargement does not result from integration of atrial cardiomyocytes into the SV, but instead develops as a result of aberrant hemodynamics. Through bulk and single cell RNA-seq, the authors characterize the cellular composition and gene expression of the main cardiac compartments (bulbus arteriosus, BA; ventricle, V; atrium, A; and sinus venosus, SV) in both wild-type controls and mutant animals. Despite certain short-comings in the analyses (see below), these profiling assays do not reveal substantial changes in the cellular compositions of BA, V and SV compartments, but instead a relative "arterialization" of the SV in *Nr2f1a*aco mutants, whereby SV transcriptomes resemble their BA counterparts more so than in controls. This intriguing observation led the authors to use bulk RNA-seq of dissected tissues and ask if the presumably equivalent blood sinuses flanking the tunicate heart also share a greater similarity with each other than with the central

myocardium.

Key merits

This focused manuscript has several key merits that warrant publication, including the characterization of the *Nf2r1aaco* phenotype, from developmental, physiological and transcriptomic perspectives, and the intriguing observations that BA and SV share under-appreciated gene expression similarities, especially in the *Nf2r1aaco* mutant, as also observed in tunicates, which hints at a possible deep origin of shared features of cardiac blood sinuses.

The study also provide rich transcriptomic datasets for future work.

General evaluation

In summary, this is an original and meritorious paper, with a suite of solid data and an intriguing hypothesis concerning the deep evolutionary origins of cardiac blood sinus. Nonetheless, several concerns remain, which warrant attention to further improve the manuscript:

Concerns

- The characterization of hypomorphic mutations could be improved, especially compared to null mutations.

The hypothesis that the *nr2f1a^{aco}* allele is hypomorphic is primarily based on the observation that there is better survival of this allele than the previously reported TALEN and CRISPR-engineered alleles (Duong et al., 2018). The engineered mutant alleles typically die by 5 days and show pleiotropic defects, including significant pericardial edema and linearized hearts. This is not observed as often with the *nr2f1a^{aco}* allele fish. Despite this, as we state in the manuscript, the *nr2f1a^{aco}* allele fails to complement the previously published engineered alleles and shows the same embryonic atrial defects. However, it is presently difficult to quantitatively show that the allele is hypomorphic, such as with Western Blots or PCR. We also recognize that there could be other alternatives to explain the differences in survivability. Regardless of the mechanism underlying the better survival of this allele, our data support this mutation is a *nr2f1a* allele that facilitated our analysis of surviving adults. Therefore, we have modified our statement in the manuscript to qualify that we think it is hypomorphic based on the survivability compared to the other alleles, but acknowledge that presently other reasons could explain this. We have also included additional data on the *nr2f1a* allele in a new supplementary figure (lines 112-121 and Supplementary Fig. 1).

- It is not entirely clear what happens to cells that would normally form the atrium in the *Nr2f1aaco* mutant. Are they integrated into the ventricle (this seems to be the case), not proliferating and/or dying altogether?

Nr2f1a mutant embryos have smaller atria, as we reported previously (Duong et al, 2018). Figure 1c and d, and Supplementary Fig. 3 show that the adult hearts still have some *Amhc⁺* cardiomyocytes, but that all *Amhc⁺* cardiomyocytes in the *nr2f1a* mutant hearts co-express the ventricular marker *Vmhc*. Additionally, our lineage tracing of embryonic atrial cardiomyocytes indicates that the adult *Amhc⁺* cardiomyocytes are derived from the embryonic atrial cardiomyocytes (Supplementary Fig. 5). However, despite the heart assuming what is essentially a single chambered heart morphology due to the almost complete loss of the atrium as we document in the manuscript, we would be hesitant to describe the remnant *Amhc-Vmhc* double positive cardiomyocytes that form a ring at the venous pole as “integrated” into the ventricle. Although the characteristic constriction at the AVC is lost, these cells still reside firmly upstream of the atrioventricular valves suggesting that they are still atrial in nature but are greatly reduced in number. We have moved the wholemount *in situ* images from the Supplement to Fig. 1c to hopefully demonstrate this more clearly.

We have not detected defects in atrial cardiomyocyte proliferation or death in the *nr2f1a* mutant embryo hearts through 96 hpf. It was not included as it is negative data and will be reported in a different manuscript examining embryonic atrial defects in the *nr2f1a* mutants. While we have not yet determined why the *Amhc*⁺/*Vmhc*⁺ cardiomyocytes population does not expand in the *nr2f1a* mutant hearts, we do not think that understanding this mechanism is critical to investigation of the current paper. We apologize if our description of the *nr2f1a* mutant atrial cardiomyocytes was not clear. We have tried to clarify statements in the text of the revised manuscript to better describe the *Amhc*⁺ cardiomyocytes in the *nr2f1a* mutants (lines 140-145).

- In Fig. 4, WT and mutant samples on same UMAP seem markedly disjointed. It is worth noting in the main text if techniques such as anchor-based alignment was performed to integrate the data from the two genotypes. The methods section makes no mention of this. In the absence of "alignment", the differential gene expression analysis may be affected. This also casts some doubts about statements in LL.170-172.

We appreciate that the reviewer highlights a common concern regarding the integration of single-cell datasets in which genetic perturbations are included. In this manuscript we include naïve integration with the software ICGS2, as opposed to joint-latent space integration, to preserve the embedding differences resulting from mutant-specific gene programs. Specifically, we find some *nr2f1a* mutant cells adopt a more arterial (bulbus) UMAP embedding for populations deriving from the venous/sinus pole. To ensure that cell cluster identity is driven by lineage as opposed to mutant-specific gene programs, we now include two orthogonal validations of the cluster predictions using independent methods. First, we perform an unsupervised projection of the *nr2f1a* mutant to the wild-type (WT) UMAP space using UMAP transform. This analysis confirmed that *nr2f1a* mutant and WT cells and corresponding cluster assignments are aligned as opposed to disjointed in the projected *nr2f1a* mutant cells, based on ICGS2 cluster assignments (Supplemental Figure 10a and b). Second, we supervised the classification of the *nr2f1a* mutant cells based on the WT assignments using the software cellHarmony. This analysis also demonstrates that the original joint unsupervised and supervised assignments are concordant (Supplemental Figure 10c). This result is as expected, as cells from no *nr2f1a* mutant or WT specific clusters were observed in the original ICGS2 analysis. Hence, the differential gene expression analysis between the WT and *nr2f1a* mutant cell populations are highly consistent and stable.

- The conclusion (LL. 175-177) should be amended as it is not clear whether the observed differences in gene expression can be attributed to the mutation, its developmental consequences and/or the differences in hemodynamics. More extensive studies with experimental design disentangling the primary developmental defects from secondary, hemodynamically-driven, defects would be needed to solve this question.

Thank you for the comment. We have modified statements within the Discussion of the revised text to clarify that we do not know if the effects on gene expression from the transcriptomics are indirect or direct (lines 496-500).

- The convergence/arterialization of SV in mutants should be substantiated by more statistically robust analyses (e.g. mutual enrichment using the detected gene set as background), in addition to the few shared markers highlighted.

As suggested by the reviewer, we have provided significantly more analyses of the bulk RNA-seq and in particular single cell RNA-seq data, which support the convergence/arterialization of cells (smooth muscle and endothelial cell) within the SV of the *nr2f1a* mutants. In the revised manuscript, we present hierarchical clustering analysis of the bulk RNA-seq data (Supplementary Fig. 4a and b and Supplementary Tables 1-4) from the zebrafish with representative markers in Fig. 1h. The arterialization

suggestion from the bulk RNA-seq is mentioned on lines 176-180 of the revised manuscript. Furthermore, in the revised manuscript, we present this observation with respect to the single cell RNA-seq data in a new subsection of the Results “Arterialization of SMCs and ECs in the adapting SV” on lines 301-374 of the revised manuscript. This deeper analysis of the data included multiple gene-set enrichment analyses comparing the arterial (BA) and venous (SV) populations as well as comparisons to a previously published single cell RNA-seq data set of zebrafish endothelial cells (Fig. 4e-j, Supplementary Figs. 15-17, and Supplementary Tables 8-19). The arterialization is also discussed in the Discussion lines 481-512.

- Likewise, the similarity between blood sinuses in *Ciona* is circumstantial in the absence of statistical analysis showing significant mutual enrichment of regional markers, and their conservation in zebrafish.

We understand this concern. We tried multiple ways to address this issue, including gene-set mutual enrichment of the *Ciona* RNA-seq data sets with the zebrafish bulk and single cell data, as was suggested. However, despite significant expertise in performing this type of analysis, we were confronted with a number of problems that prevented us from getting a result with an unbiased statistical approach. The primary issue we were confronted with is that the *Ciona* genome is still very poorly annotated. Consequently, our *Ciona* gene sets despite quality sequencing were very limited in number. While homologs of the genes appear to be conserved in the blood sinuses (BSs) and BA/SV between *Ciona* and zebrafish via manual annotation, typically there are many more homologous genes in zebrafish compared to single homologs in *Ciona*. This is understandable due to the genome duplication events in vertebrates and teleosts. Even if we tried including all homologous genes in zebrafish for the gene-set enrichment, the issue remained the limited number of genes identified in our conditions to start with due to the currently available *Ciona* genome. Therefore, we regret that we presently cannot perform this type of unbiased mutual enrichment analysis comparing the *Ciona* and zebrafish samples, despite considerable effort. We have tried to modify how we phrase our interpretations to acknowledge this.

Please also recognize that in the revised manuscript we have re-analyzed most of our data and now include hierarchical clustering of the *Ciona* data with an additional biological replicate (lines 376-415 and Fig. 5b, Supplementary Fig. 19, and Supplementary Tables 20 and 21). In doing this analysis, we inexplicably have found that a lot of the annotation of the *Ciona* genome that was available seems to have changed compared to our initial analysis presented in the original manuscript. While the data support the same fundamental observations, we have updated the manuscript accordingly and changes in interpretations are reflected in the revised manuscript.

- The orthology assignment between *Ciona* and zebrafish genes are not explained. This is important to conclude about conserved regulatory programs for

As stated above, *Ciona* typically have single homologous genes for multiple vertebrate/zebrafish genes. To confirm orthology/homology between genes, the orthologous gene names for *Ciona* were assigned following the RNA-seq analysis using the Gene ID conversion tool from the DAVID website (<https://david.ncifcrf.gov/conversion.jsp>), the tools in the BioMart for assigning orthology on the Ensembl website (ensembl.org), and for some the Aniseed tunicate website (aniseed.fr). Hence, this was an unbiased method to assign basic homology/orthology to the genes found in *Ciona* and zebrafish. In a few cases, *Ciona* and zebrafish orthologues were not found to have similar expression with the BA and SV. However, some homologous gene family members were, which were included.

- Discussion of smooth muscle vs endothelial cells, and potential pacemaker cells at either side of the *Ciona* heart tube are worth discussing..

Thank you. We have significantly expanded the Discussion of the composition and evolution of the *Ciona* BSs and how this relates to the evolutionary hypotheses within the revised manuscript (lines 514-553).

- fig.1f shows similarity only for a few genes. A PCA analysis (as shown in Ext Data Fig. 7, which could be moved to the main figure) PC plots of samples would be more convincing and/or biclustering analysis of conditions and genes

As suggested, we have included PCA for the zebrafish bulk RNA-seq analysis in Fig. 1 of the revised manuscript. It is now referred to on lines 155-160 and present in Fig. 1g of the revised manuscript. The heatmap presented in Fig. 1g is representative genes taken from the clustering analysis presented in Supplementary Fig. 4a.

We have also included PCA and hierarchical clustering for the *Ciona* data in the revised manuscript.

- Fig.1j. Indicate marker used for neural crest, not NC:mCherry

The term *NC:mCherry* refers to the combination of transgenes *Tg(sox10:GAL4,UAS:Cre)* and *Tg(-3.5ubb:loxP-EGFP-loxP-mCherry)* used to label neural crest-derived cells. We used the name *NC:mCherry* to simplify the designation of the combination of these transgenes as was done when it was published (Cavanaugh et al, 2015). We apologize for the confusion and recognize this designation was not clearly defined in the original manuscript. We have defined *NC:mCherry* in the revised manuscript (lines 186-190).

Reviewer #2 (Remarks to the Author):

Adaptation of the sinus venosus reveals insights into evolutionary transitions of the vertebrate heart by Gafranek et al.

In this study, nr2f1a hypomorphic mutants are being investigated. Like nr2f1a mutants, these fish show atrial dysmorphogenesis, but a fraction of the fish survive and have been studied here. The fish show absence of atrial muscle, and an enlarged sinus venosus seems to be connected to the ventricle. Transcriptomes of mutant sinus venosus “replacing” the atrial compartment resembles both the sinus venosus of wild type fish and, to some extent, the outflow tract (bulbus arteriosus). The mutant sinus venosus is subject to retrograde blood flow, and altering sheer stress impacted on wall thickness. Single cell RNA sequencing suggested that mutant sinus venosus smooth muscle cells had differentiation state similar to those of both wt and mutant bulbus arteriosus. They also compared the transcriptomes with those of *Ciona*, and found indications supporting the ancestral equivalency between the large vessels that flank the vertebrate heart. The authors claim their “data reveal that the foundation of a rudimentary chordate heart with equivalent flanking vessel-like chambers remains latent within basal vertebrates, and hint at evolutionary origins underlying vascular disease”.

Comments

1) An important assumption underlying this paper is that the sinus venosus of the zebrafish heart represents that of vertebrates, and therefore can be used as model to gain insight into evolutionary origins. However, the sinus venosus of zebrafish is vascular, whereas the sinus venosus of most other vertebrate species investigated is cardio-muscular. Therefore, the zebrafish sinus venosus can be considered to be specialized, in that respect not representing the sinus venosus most commonly encountered in vertebrate species. This derived state of the sinus venosus makes any comparison with “ancestral states” found in *Ciona* difficult at best. In fact, the authors themselves raise this point in line 81, but do not take into account the implication for their analysis and conclusions.

a. We appreciate this comment. However, we do not think it is not a completely accurate characterization of what we stated in the original manuscript regarding the evolution of the SV. We do not make assumptions that the SV of the zebrafish heart represents that of all vertebrates. It can be stated that the SV of all vertebrates has some cardiomyocytes (Jensen et al. 2014; Carmona et al. 2018). However, the amount of the cardiac muscle within vertebrate SV can seemingly vary quite a bit, which is acknowledged though not really depicted in Jensen et al, 2014. Teleosts may have less than others (Jensen et al. 2014; Tessadori et al. 2012; Farrell and Smith 2017), but they still have some overlap of cardiac and smooth muscle at the border between the atrium and SV. In the revised manuscript, we show this overlap with immunohistochemistry in Supplemental Fig. 21. However, based on the current literature, we find it difficult to make definitive conclusions about the amount and types of cardiac muscle, as well as other cell types, within the SV, as virtually all the characterization has been histological and lacks molecular markers and anatomical standardization (Yamauchi 1980; Robert M. Santer 1985; R. M. Santer and Cobb 1972; Satchell 1991). Moreover, if there are cardiomyocytes within the SV of all vertebrates and some authors are defining the SV as a cardiomyocyte-containing region adjacent to the venous atrium, then it's possible that comparable regions of the vertebrate hearts are not properly defined. Nevertheless, what is currently defined as the SV in all organisms is also comprised of smooth muscle and endothelial cells too. We do not want to weigh down the current manuscript with semantic arguments focused on redefining comparable segments of vertebrate inflow anatomy. Minimally, we think that to better inform evolutionary hypotheses of the vertebrate SV, the composition of the vertebrate SV in representative vertebrates need to be verified with cell-type specific molecular markers coupled with anatomical references.

b. Even if we make the assumption that the SV in zebrafish is somewhat specialized with respect to other vertebrates, this does not preclude that within teleosts the SV has reverted to a more ancestral-like state, particularly when compared to tunicate hearts. Evolution does not always reflect a seemingly parsimonious advancement of traits. Hence, one possibility is that the composition of the teleost SV has reverted to be more representative of a fundamental plan and transition state that is similar to an ancestral early-branching chordate hearts with a cardiac chamber flanked by two more distinct vascular vessels. However, the presence of some putative striated muscle in the agnathan SV does not mean it is a homologous structure to what is found in later-branching vertebrates. There are anatomical differences in the agnathan hearts and SV compared to chondrichthyan, teleosts, and other vertebrates. Additionally, the electrophysiology referenced in Jensen et al, 2014 is not consistent throughout all agnathans. More recent studies have not been able to detect discernible activation waves in the SV of other hagfish species (Icardo et al. 2016). Therefore, these observations, as well as additional specializations found in agnathan hearts, have caused some in the literature to question whether or not agnathans are good representatives of ancestral traits within the hearts of early-branching vertebrates and have even suggested that investigative efforts in the field are better expended exploring other species (Icardo et al. 2016; Icardo 2017). As to fully represent these different evolutionary hypotheses, we now present all of these ideas in the expanded Discussion of the revised manuscript (lines 554-582).

c. We recognize that because of the short format of our original manuscript that we did not explicitly acknowledge and thoroughly discuss some of these more nuanced evolutionary arguments in the literature and discuss how our data expand this previous work. In the revised manuscript, we have significantly expanded the Discussion to acknowledge and discuss these different ideas and how our data relate to evolutionary hypotheses of the SV (lines 554-609).

2) To me it is not clear how the atrial muscle has disappeared in the mature mutant fish. Is it not formed, the sinus venosus then by definition connected to the ventricle? Figure 1a suggests an atrial compartment-like structure is formed, but devoid of cardiomyocytes. Then, did the atrial cardiomyocytes disappear? In that case, the cardiomyocytes may have "simply" be replaced by fibroblasts and myofibroblasts and smooth muscle, not unlike what is seen in atrial fibrillation and aging. The "atrial" compartment may now adopt a vascular phenotype, and called "sinus venosus" by the authors. Or is it enlargement of the sinus venosus compensating for the (gradual) disappearance of the atrial compartment? Please clarify.

a. These comments seem to indicate some confusion that does not accurately reflect what we have shown in the manuscript. As we published (Duong et al, 2018 and Dohn et al, 2019), *nr2f1a* mutant embryos have smaller atria. Our data in the present manuscript show that *Amhc*⁺ cardiomyocytes do not completely disappear and those that remain in adults are derived from the embryonic *Amhc*⁺ (atrial) cardiomyocytes.

b. Figure 1a shows that an enlarged structure is at the venous pole of the *nr2f1a* mutant hearts. As stated in the text, we initially presumed this structure was the “atria” of the *nr2f1a* mutants. It is marked with “A” in the text and for Fig. 1a and b to indicate to the reader our initial lack of understanding of what the enlarged structure actually is derived from at the venous pole prior to our marker analysis that subsequently showed this enlarged structure at the venous pole is the SV and not atrial in nature. Figure 1c-e, and Supplementary Fig. 2 show that the adult hearts still have some *Amhc*⁺ cardiomyocytes at the venous pole. However, this cardiomyocyte population is significantly reduced in the *nr2f1a* mutants, such that is not a discrete chamber, and all *Amhc*⁺ cardiomyocytes co-express the ventricular marker *Vmhc*. Therefore, in the adult *nr2f1a* mutant hearts, the enlarged SV is adjacent to a smaller group of cardiomyocytes that essentially form a ring at the venous pole of the single chambered heart and co-express *Amhc* and *Vmhc*.

We have not yet established why the *Amhc*⁺/*Vmhc*⁺ cardiomyocyte population in the *nr2f1a* mutant hearts does not expand. We hypothesize it is due to a lack of proliferation, as lineage tracing experiments of *Amhc*⁺ atrial cardiomyocytes in Supplementary Fig. 5a and b of the revised manuscript shows that the embryonic *Amhc*⁺ cardiomyocytes in the embryo gives rise to *Amhc*⁺ cardiomyocytes in adults, suggesting there is not significant death of the embryonic *Amhc*⁺ cardiomyocytes. However, understanding specifically why this population of cardiomyocytes does not expand in the *nr2f1a* mutants is not critical to this paper.

c. We have no evidence from our data that a previously existing atrial chamber is at some point replaced by fibroblasts or myofibroblasts and adapting a vascular phenotype. If there were infiltrating cells contributing to the enlargement of the SV, these would have been evident from the scRNA-seq and our histological analysis. We state that cell infiltration was not observed in the original manuscript (lines 219-220 of the revised manuscript). Hence, our interpretation is in line with the last statement from this comment: We posit that the enlargement/adaptation of the SV is due to aberrant blood flow, which is likely the result of the lack of a true atrium. While it is a possibility, it is not clear that “compensating” is the correct term for what is happening to the SV without additional evidence. We did not use that specific term in the manuscript to describe what is happening in this context.

d. We apologize that our description of what is happening to the atrium/atrial cardiomyocytes in the *nr2f1a* mutant hearts was not clear in the original manuscript. We have tried to clarify statements in the text of the revised manuscript to make our description of the atrium and SV in mutants clear (lines 125-131 and 140-145). Also see responses to Reviewer #1’s comments 1 and 2.

3) The authors seem to push the idea that the sinus venosus of the mutant adopts a phenotype resembling that of the bulbus arteriosus. Most comparisons in this study take into account a few specific transcripts that would show resemblance between mutant SV and the BA (e.g. fig 1f, etc). The selection of genes does not represent the actual transcriptomes, and should be replaced by a hypothesis free unbiased analysis of representative transcriptomes. Indeed, the PC analysis of the RNAseq data and the sc-RNAseq data indicates that the mutant sinus venosus mostly resembles wt sinus venosus, with only selected markers indicating resemblance with bulbus arteriosus. In fact, figure 4c indicates a larger difference between wt and mutant bulbus arteriosus.

a. With respect to Fig. 1f of the original manuscript, those data alone were not meant to make broader statements about the *nr2f1a* mutant SV adopting more of a resemblance to the BA. To address this comment, in the revised manuscript, we have included unbiased analysis of the zebrafish bulk RNA-seq

data set. While these data of course show gene expression differences in the BA and SV, they also show many similarities in gene expression, including enrichment for the SMC markers. The PCA for the bulk RNA-seq is now presented in Fig. 1g. The overall hierarchical clustering is presented in Supplementary Fig. 4a and b of the revised manuscript. These data are referred to on lines 155-160 of the revised manuscript. Fig. 1h of the revised manuscript now includes representative markers for different cell types taken from the hierarchical clustering analysis and is discussed on lines 160-168. The heatmap in Fig. 1h shows that the *nr2f1a* mutant SV and the WT SV express canonical SMC markers, which are also expressed in the WT BA, as well as arterial and venous EC markers. The list of genes that shared expression in the WT BA and SV is presented in Supplementary Table 4. The composition of smooth muscle in the SV of WT and *nr2f1a* mutant fish was subsequently confirmed with immunohistochemistry (Fig. j-n and Supplementary Fig. 4d). Importantly, this data indicates that the enlarged venous structure in the *nr2f1a* mutants is derived from the SV, not the atrium, and that the WT SV has cells that express canonical smooth muscle markers. The molecular and cellular composition of the SV has not been previously examined in zebrafish (or any fish to our knowledge) and importantly these SMC markers were previously stated to be specific to the BA. We think it is appropriate to show a selection of established marker genes to make this point in the main section of the manuscript. Presenting a selection of representative marker genes from larger sequencing data sets in this way is common within the literature and consistent with standards in the field (Dobnikar et al. 2018; Mantri et al. 2021; Miklas et al. 2022; Sánchez-Iranzo et al. 2018). The larger data set and clustering analysis is presented in the Supplement.

b. Yes, our interpretations of the data are that the SV is coming to resemble the BA. These interpretations are based on the following observations: 1. Morphologically the SV in *nr2f1a* mutants becomes enlarged with thicker walls (Fig. 1, Fig. 2, and Supplementary Fig. 6); 2. The morphological changes correlate with PCA from the bulk RNA-seq data set (Fig. 1g of the revised manuscript); 3. The morphological changes correlate with the scRNA-seq data showing that molecularly and cellularly subpopulations of SMCs and ECs from the *nr2f1a* mutant SV are taking on arterial identity. In the revised manuscript, we have significantly expanded our analysis of the gene expression and cell types within the BA and SV, which further bolster the hypothesis that these cells within the SV are adopting arterial identity. These data are presented in Fig. 1h and Supplementary Fig. 4a and b for the bulk RNA-seq and Fig. 4e-j and Supplementary Figs. 15-17 for the single cell RNA-seq. The data are discussed on lines 155-160 and 301-355 of the Results and lines 481-512 of the Discussion of the revised manuscript.

c. We agree that a hypothesis suggested from the UMAP data of the single cell RNA-seq analysis could be that arterial SMCs in the mutant *nr2f1a* BA (C1) are becoming more “venous-like” based on the intermediate position between WT arterial SMC cluster (C2) and venous SMC cluster (C6). In the revised manuscript, we have performed specific comparisons of the C1 and C2 populations, including gene-set enrichment analysis. Our data show that the major decreases in gene expression in the *nr2f1a* mutant BA vs WT and C1 vs C2 SMC clusters are associated contractility and metabolism, and are not changes to a more “venous-like” identity of SMCs. Therefore, the change in location within the UMAP is consistent with the hypothesis that the BA in *nr2f1a* mutants exhibits secondary changes due to experiencing reduced flow and shear stress, while the SV at the venous pole experiences increased shear stress. However, in considering this interpretation from the UMAP of the *nr2f1a* mutant BA, it is worth noting that it is primarily the arterial SMC cluster (C1) that shows this positioning. Despite the positions within the UMAP, the differentiation state analysis of the SMC populations supports a trend that the WT and *nr2f1a* mutant BA SMCs populations remain pretty similar (Fig. 4i-j). In the revised manuscript, the comparison of the BA SMC populations is presented on lines 356-374 of the Results.

Therefore, we have performed unbiased analyses of the transcriptomes, which support that subpopulations of the venous SMC and ECs in the *nr2f1a* mutants are adopting arterial-like identity and differences in the arterial SMC populations are likely due to indirect effects from aberrant blood flow and shear stress.

4) Line 105 (p5). This is incorrect. Atrial cells do not invade the vascular venous pole. The atria and subsequently the myocardial sinus venosus are formed from caudal second heart field progenitors in

chronological order. Mouse lineage studies have shown that once differentiated to atrial muscle, these cells do not invade sinus venosus, and sinus venosus myocardium does not invade the venous vasculature.

We agree that the statement we used in the original manuscript was not accurate. We were referring to the “atrialization” of this tissue, such as from Jensen et al. 2014, and that the SV and atria in mammalian embryos is more distinct and then incorporates with the venous atrium (Jensen et al. 2014; 2017; Faber et al. 2019; Carmona et al. 2018). We have removed the statement and modified the text to improve the presentation of the rationale for this experiment (lines 196-202).

5) Lines 175-177. This seems to be an overinterpretation. Did the authors study molecular identity in response to aberrant hemodynamic forces? These data are not provided as far as I can see.

Our data show that there are aberrant hemodynamic forces within the *nr2f1a* mutant hearts. It is not clear how one would directly study molecular changes due to these hemodynamic forces in this context. However, the cellular and molecular changes we observe correlate with the aberrant hemodynamic forces in the *nr2f1a* mutant hearts. The statement indicated in the comment referred to the arterial-like SMC and EC populations found within the *nr2f1a* mutant SV, which correlates with these aberrant hemodynamic forces. We have added the in-depth comparisons of cell types within the BA and SV indicated in the previous comments and have modified this statement in the Results of the revised manuscript to clarify there is an association between the changes of the cells types in the SV and the aberrant hemodynamic forces found in *nr2f1a* mutant hearts (lines 371-374).

6) An issue that the authors may want to address: It is possible that the changed flow impacts on the vascular sinus venosus of mutants, thereby imposing some outflow (arterial) vessel-like phenotypic aspects. This is an indirect consequence of AV valve dysfunction (I assume). Shear stress effects on vascular (arterial/venous) tissue have been well documented. But how does this finding have implications for atrial/sinus venosus evolution?

Mutants in zebrafish, and other vertebrate models, can be used to provide information about both diseases in humans and animal evolution. It has been proposed that tunicate hearts reflect an ancestral heart that beats bidirectionally, suggesting a hypothesis that this heart may lack asymmetry found at their arterial and venous poles. Hence, one might predict there would be greater similarity in gene expression between the arterial/pharyngeal and venous/stomach blood sinuses (BSs), since they can both handle pressure from blood being expelled at either pole of the heart. Our *Ciona* expression data indeed, minimally, suggest that tunicate hearts seem to express orthologs of markers found in the SMCs and ECs of the BA and SV in fish in both BSs. Furthermore, tunicate orthologs of vertebrate heart markers that show chamber-specific expression appear to be expressed in the single cardiac chamber. Thus, these data support that tunicate heart has greater symmetry in gene expression in their arterial and venous BSs.

Our data have implications for understanding atrial/SV evolution because essentially the *nr2f1a* mutants do not have an atrial chamber, which we posit reflects a more ancestral-like heart. As we show in the manuscript and is described in our responses to comments above, the venous pole of the zebrafish heart in the *nr2f1a* mutant hearts comes to also resemble aspects of the arterial pole, which correlates with the aberrant retrograde blood flow. The evolutionary comparisons suggest a hypothesis whereby these auxiliary vessels may have become more specialized with the appearance of unidirectional flow from the appearance of multiple chambers, yet in the absence of an atrium there remains the latent ability to revert to a more ancestral like heart with less polarity and specialization at the arterial and venous poles. This hypothesis is discussed in greater detail in the Discussion of the revised manuscript (509-548).

7) The value and implications of the comparison with *Ciona* are not clear to me. Selected genes point to resemblance of vascular poles of *Ciona* and zebrafish. Indeed, both species have vascular poles,

showing resemblance. How is that novel, and how does that provide insight into evolutionary relationships? (see also point 1)

The data are novel because there is not much known about the molecular nature of the BSs in tunicates or the molecular composition of the zebrafish SV and they have not been compared previously. Our data suggest the BSs express genes expressed in SMCs and ECs of vertebrates, providing new insight into gene expression within the tunicate BSs flanking the heart. From a comparative perspective, there is also limited, if any, comparison to the arterial and venous tissues flanking the hearts in zebrafish or other vertebrates to the tunicate BSs. Indeed, both hearts have vascular poles, although whether or not there are true SMCs in tunicates need further investigation, that show resemblance is an important observation for anatomical comparison and supports new hypotheses about the origins and evolutionary transitions of the auxiliary vascular chambers flanking vertebrate hearts. This point is also addressed in the revised manuscript (lines 514-553).

8) With the loss of the atrial compartment, the AV canal myocardium (positive for a and v myosin) seems to be expanded in mutants. Please comment on this.

As we reported previously in Duong et al (2018), the AV canal is expanded in *nr2f1a* mutant embryos as defined by the overlap of *Amhc* and *Vmhc* markers. As stated above in response to comment 2, in the adult *nr2f1a* mutant hearts, it appears that all the *Amhc*⁺ cardiomyocytes come to co-express *Vmhc* (Fig. 1d,e and Supplementary Fig. 3). While some of these cardiomyocytes likely represent part of the expanded AV canal, the location of the *Amhc*⁺/*Vmhc*⁺ cardiomyocytes would suggest there is a relatively small amount of cells that extend past the AV canal toward the SV. However, it is hard to define given the aberrant morphology of the heart and the remaining *Amhc*⁺/*Vmhc*⁺ cardiomyocytes at the venous pole of the *nr2f1a* mutant hearts.

9) An alternative explanation for authors findings is that *Nr2f1a* imposes venous identity by directly regulating Notch signaling. In Nature 2005, You et al (PMID: 15875024) observed in mice: “Ablation of COUP-TFII in endothelial cells enables veins to acquire arterial characteristics”. Something very similar may be happening in the mutant fish sinus venosus. This provides a good explanation for many of the observations by the authors. The link between venous pole (including atrial) identity and evolution and *Nr2f*/Coups-TF has been implicated already in literature. It remains unclear to me how authors findings provide more insight into venous pole evolution, and the implications for vascular disease seem unsupported by the provided data.

a. Yes, it has been well documented that *Nr2f* factors can promote venous EC identity through inhibiting Notch (You et al, 2005). While aspects of our data may seem similar, it is not clear to us that this data provides a good explanation for many of the observations in the manuscript based on the following: 1. Within our data set, we do not observe a complete transformation of the venous SMC and EC populations. It is only subsets of cells within the *nr2f1a* mutant SV that exhibit “arterial” identity. (Please see response to comment 3 for additional details); 2. We are not aware that it has been shown previously that *Nr2f* factors promote venous vascular SMC identity and repress arterial SMC identity; 3. Given the relative lateness of the SV adaptation, we presently cannot distinguish if these fate transformations are directly due to loss of *Nr2f1a* or are secondary due to adaptation from the improper hemodynamic forces. We acknowledge this on lines 496-500 of the revised manuscript; 4. We do not observe defects in Notch signaling, differential gene expression analysis of our transcriptomic data (Supplementary Fig. 15), nor within the *nr2f1a* mutant embryos do we observe gross defects in vascular patterning (Supplementary Fig. 20) or either from a transgenic Notch reporter line (not shown). We directly refer to these new data in the revised Discussion (lines 481-512).

b. Regarding the comments about evolution and disease, it is not clear to us what Reviewer 2 thinks has already been demonstrated in the literature about the evolutionary role of *Nr2f* factors in heart evolution. We disagree that our data do not use a new model to significantly advance our understanding of vertebrate venous pole evolution and model cardiovascular disease. This comment from Reviewer 2

contrasts significantly with those from Reviewer 1. Briefly, our data provide novel insights into the composition of the SV and consequences of changes from loss of Nr2f1a that can be used to provide insights into patterning of the heart. In turn, with respect to heart evolution, these observations can be used for comparative analysis and evolutionary inferences about transition states of the heart and its auxiliary vessels. Moreover, our data simultaneously provide a unique zebrafish model for studying congenital heart defects in vertebrate embryos and adults, paralleling the long-term progression of cardiovascular disease found in adults patients with structural congenital heart defects, which cannot be done in global murine *Nr2f2* knockouts as they are embryonic lethal. Specifically, our data show that the adaptations of the *nr2f1a* mutant SV exhibit secondary consequences that reflect a conserved vascular hypertensive remodeling response, similar to what is found in surviving patients with structural heart defects. We have dramatically expanded our Discussion within the revised manuscript to try to address and clarify each of these points.

Reviewer #3 (Remarks to the Author):

General comments

The study by Grafanek et al reports on the phenotype in zebrafish of loss of nr2f1a (Coup-tfII in mammals) translation, with approximately half of the fish living to adulthood. In the dying fish, there are several signs of cardiovascular disease, but we are not told whether the loss of nr2f1a has effects outside of the heart. In mouse, loss of Coup-tfII has pronounced effects on the extracardiac vasculature. While there is a clear cardiac phenotype, such as absence of myocardium in the atrium, we do not gain much insight to what extent the vasculature is dysfunctional and the cause and effect of the pathology is quite obscure. Atrioventricular valve insufficiency is clearly documented. An attempt to link this to altered shear stress is made by estradiol treatment which associates with vasodilation. The experiment, however, is at best very indirect since this steroid hormone has profound transcriptional effects in a broad range of tissues. Bulk and single cell RNA-sequencing is used to compare different compartments of the heart of WT and nr2f1a mutant fish, but it is a major short-coming to these experiments that no systemic vessels are included as benchmark reference. This in turn obscures the stated aim of the study which is to gain insight into how the chambered heart evolved. It may indeed be possible to unmask ancestral gene programs in zebrafish (Hawkins et al 2021 Cell). The present manuscript, however, does not have a gold-standard reference to what could be unmasked and a dysfunctional heart does not make a compelling case. Many of the reported experiments have short-comings in design and, or, analysis – please see specific points below – and so while in aggregate an appreciable amount of work has been done, the manuscript at present does not have a core of solid findings.

Thank you for the general comments. However, we do not think that some of the Reviewer #3's comments accurately reflect statements we made in the original manuscript.

a. To be clear, we never state the *nr2f1a* mutant fish are dying fish. This seems to be an inference made by Reviewer 3. As stated in the text (lines 121-123), some of the homozygous *nr2f1a* mutant fish die as we do not have 100% viability, though this is likely not from the heart defects. However, *nr2f1a* mutant fish that survive to adulthood, even with ostensible edema and cardiac enlargement, can live for some time and perform well in functional tests, such as the swim tunnel (Supplementary Fig. 8).

b. We agree that the estradiol experiment may have indirect effects. However, we disagree that even if the method of action is indirect that this is a problem with the interpretation and that there would be better, feasible ways to test this hypothesis experimentally. Pharmacological vasodilators are commonly used in the clinic to treat hypertension and experimental models of hypertension, regardless of the directness of action. Minimally, our data with estradiol, which is used to treat animal hypertension models, supports the adaptive thickening of the SV is a hypertensive mechanism. Additional pharmacological vasodilators

that act through a different mechanism still show the same effect as estradiol, corroborating the adaptive response is a hypertensive mechanism irrespective of the directness of action. The use of these additional vasodilators has been added to the revised manuscript (lines 231-241, Figure 3b,c and Supplemental Figure 7a-d). Please see response below for additional comments.

Major points

1. This study on the evolution of the vertebrate heart has a well-argued premise, namely that there is a dearth in hearts that could represent transitional stages between the looped heart tube of sea squirts and the chambered hearts of vertebrates. It is a simplification, however, to reduce the heart chambers of agnathans to comprise an atrium and a ventricle only, since the sinus venosus of these hearts also contain myocardium and have distinct deflections on the ECG. Confusingly, you acknowledge this on L80-81 and correctly you are citing your reference 23 to support this. The evolutionary origins of the bulbus arteriosus is debated, namely whether it evolved in cartilaginous fishes or in teleost fish (e.g. Moriyama et al versus Lorenzale et al), but I do not know of any study that argues that the agnathan outflow tract is homologous to the bulbus arteriosus. My major concern is then that the study has failed to realise that the teleost heart is highly specialized by the absence of, or at least extreme reduction in myocardium, in the sinus venosus and the outflow tract.

a. We appreciate that Reviewer #3 states that we have a “well-argued” premise. However, it is not clear to us that other statements in this comment reflect what we stated in the original manuscript. We did not state the agnathan heart is only comprised of an atrium and ventricle. However, this has actually been proposed in the literature (Icardo 2017; Icardo et al. 2016). We acknowledge it has been reported there is some striated muscle in the SV of agnathans, as was cited in Jensen et al, 2014, and that this has been interpreted as an ancestral trait of the SV (Jensen et al, 2014). These were points that were likely not made clear due to the short format of the original manuscript. To clarify our interpretations, we have significantly expanded the Discussion of the agnathan heart data in the revised manuscript (lines 554-609). However, in evaluating the composition and extent of proposed cardiomyocytes in the agnathan SV, as well as the SV throughout vertebrates, the histological analysis upon which these are based that is currently in the literature needs to be updated with molecular characterization and standardization of immunohistology within the hearts, so that the type, extent, and location of the cell types within the SV of different vertebrates is understood for proper comparisons. Additionally, in examining the reported ECG data, the size of the inflection is minimal, suggesting that the amount of cardiomyocytes with the agnathan SV is also minimal, at best. It is also difficult to interpret whether there is significant if any functional contribution given how small the proposed SV wave is in agnathans, or really any fish. It is also worth noting that other groups have not been able to detect nodal activity in agnathan SVs save for a single species of hagfish (Icardo et al. 2016). Additionally, despite what is proposed in Jensen et al., 2014, it is not agreed upon that the agnathan SV is homologous to other vertebrates. Because of these observations, it has even been proposed that agnathans be abandoned for the purposes of drawing evolutionary conclusions regarding the basal vertebrate inflow tract (Icardo et al. 2016). Again, we have expanded our Discussion of these data and interpretations to include these conflicting hypotheses in the revised manuscript (lines 554-609).

b. With respect to the origins of BA and SV, we did not make statements about the agnathan outflow tract being homologous to the BA. However, there is strong anatomical and molecular data supporting cartilaginous fish and teleosts both having a BA and conus arteriosus (CA) (Durán et al. 2008; Lorenzale et al. 2018), although the conus arteriosus is significantly reduced in teleosts (Icardo 2006; A. C. Grimes and Kirby 2009; Jones and Braun 2011). Our data support that the *Ciona* BSs have less polarity and hence greater equivalency with respect to certain identifiable markers. We make the observations that the zebrafish BA and SV have SMCs, that the venous ECs express markers that are typically thought to be arterial, and that the *nr2f1a* mutant causes a state that reflects a vertebrate heart lacking an atrium.

We have expanded the Discussion of our data to include these interpretations in relation to the literature (respectively - lines 441-462, 463-480, 545-553).

c. Regarding the teleost heart, we acknowledge the possibility that it may be more of a specialized state compared to some vertebrates. However, it has also been proposed that the agnathan heart may be more derived, with what has been called the SV in agnathans not actually being homologous to the SV in other later-branching vertebrates. Therefore, the agnathan SV may not be reflective of more ancient vertebrate hearts (Icardo et al. 2016; Icardo 2017), or at least closer to a transitional state between urochordate-like ancestors and vertebrate hearts. The amount of striated muscle within the SVs throughout vertebrates may also be more of a semantic argument, as they all contain muscle to varying degrees, it just appears that a transition zone harboring cardiomyocytes and smooth muscle is reduced in teleosts. Nevertheless, whether or not the teleost/zebrafish BA and SV are derived or more specialized does not rule out that they originated from BS-like structures in an ancestral chordate heart and that they currently may reflect a more ancestral state where there was less muscularization at the arterial and venous poles. We have expanded the Discussion to address these points (lines 554-609).

2. Your use of the term 'chamber' is without definition and it is clearly at odds with a very large fraction of the literature. For example, a chamber should have myocardium and have electrical activity that is distinct in time from that of the other chambers. Defined such, the sinus venosus of all fish have a sinus venosus chamber, except some teleosts! On the other hand, the bulbus arteriosus, when defined as the bulb-shaped structure that gives rise to the name of it, does not have myocardium and it is without the distinct electrical activation and repolarization seen in chambers, so it is not a cardiac chamber but an arterial structure.

We understand "chamber" from a biological and anatomical definition in Merriam-Webster's medical dictionary to mean "an enclosed space in the body of an organism; a cavity." There are "chambers" in other organs besides the myocardial part of the heart within vertebrate bodies, such as the vitreous chamber behind the eye. Hence, using the term "chamber" alone within the literature does not strictly mean a myocardial cavity with electrical activity in the heart. This definition of "chamber" may be inferred in some cardiovascular-related literature and not need greater definition. However, this is not the case more broadly. Moreover, within the literature, the zebrafish/teleost heart has often and commonly been referred to as having 4 chambers comprised of the BA, V, A, and SV (Adrian C. Grimes et al. 2006; Tessadori et al. 2012; Simões et al. 2002; Stainier, Lee, and Fishman 1993; Hu et al. 2000; Hu, Yost, and Clark 2001; Satchell 1991; Robert M. Santer 1985). Our use of the term "chamber" or "chamber-like" was in reference to the structures being ostensibly large, enclosed spaces or cavities from published and work shown in the manuscript. Throughout the revised manuscript we have tried to make sure we defined whether or not the "chambers" being referenced were cardiac or vascular in nature.

3. L78-80. I fail to see the logic here. Why would the loss of atrial identity / chamber formation, promote sinus venosus formation? It is perhaps an interesting hypothesis that the components upstream of the ventricle were vascular in a hypothetical ventricular/one-chambered heart evolutionary stage and that the atrial identity has been imposed on top of that. If so, you could articulate this, but other hypotheses could also explain the appearance of more than one chamber.

We apologize for any confusion that could have been caused by that statement. With the statement in the original manuscript, "*Given its anatomical position, we reasoned that the enlarged, dysmorphic structure that replaced the atria in *nr2f1a^{aco}* mutant hearts may be part of, or derived from, the venous inflow vasculature, and specifically the sinus venosus (SV),*" we did not mean that the loss of atrial chamber identity would promote SV formation in the mutants. This statement was in reference to understanding what the dysmorphic chamber was originating from. We initially thought it was part of or derived from the atrium in the *nr2f1a* mutants, due to its overt appearance. However, clearly all our transcriptomic and immunohistochemical analysis suggested it was not. The statement refers to the hypothesis that the enlarged structure that is in place of the atrium in *nr2f1a* mutants was actually derived

from the SV. We have revised this sentence and other points within the paragraph to clarify our rationale for these experiments in the revised manuscript (lines 146-150).

4. Your bulk RNAseq is indicative of similarity between the vascular/non-chamber sinus venosus and the bulbus arteriosus. This similarity may seem striking in comparisons to the very different tissues of actual chambers (atrium and ventricle), but in fact proper reference tissue have not been included, say a large artery and a large vein. Was the mutant bulbus arteriosus not included (Figure 1f) and if so, why not? Fig 1f: Why were only these 10 genes specifically selected for analysis? How significant is their differential expression in mutant tissues? Overview of what this mutation does to the atrial/SV/BA/etc transcriptome, GO-term analysis?

a. For the bulk RNA-seq, there were not comparable data sets from adult zebrafish to use to compare to a large artery or vein. Therefore, we have performed gene-set enrichment comparisons using our single cell RNA-seq EC clusters with those from single cell RNA-seq analysis of the ECs in the embryonic zebrafish, which identified multiple different EC populations. This data confirms the arterial and venous nature of the ECs from the adult BA and SV, but also demonstrates the venous ECs in the SV have similarity in gene expression to an intermediate vascular EC population that senses pressure. In the revised manuscript, this comparison is included on lines 278-299 and the data are presented in Fig. 4d,e and Supplementary Tables 6,7,13, and 14.

b. The BA was not included for the *nr2f1a* mutants because when we initially performed the experiment, we did not think it would be informative. Given we then chose to perform single cell RNA-seq analysis of the WT and *nr2f1a* mutant SV, we reasoned that it would not be as informative or change the interpretations if we were to go back and assess the *nr2f1a* mutant BA with bulk RNA-seq.

c. We chose representative genes for Fig. 1f of the original manuscript that show the SV expresses smooth muscle markers. In the revised manuscript, we have provided clustering analysis (Supplementary Fig. 4a,b). Fig. 1h of the revised manuscript has select SMC and EC markers from these datasets. The intent is to show the smooth muscle gene expression in the SV, which was not previously reported. This data is now described on lines 155-168. Additionally, please see our response a to Reviewer #2, comment 3.

d. The differential expression analysis and Z-scores for the zebrafish BA, SV, V, and A bulk RNA-seq analysis are included in Supplementary Tables 1-3. We did not present GO-terms for this analysis due to the size of the manuscript and that it did not aid in the interpretations for those figures. We have included the GO-term analysis of scRNA-seq comparisons (Supplementary Tables 10, 11, 13, 16, 19).

5. L104-110. I don't understand on what grounds/published data that you would hypothesize that atrial tissue contributes to the SV - and your data also bears out that this hypothesis is ill-founded. Also, given point 4 above, it is not clear to me that this part informs us on the role of the non-myocardial sinus venosus in contributing to the aberrant atrial segment.

We apologize that this statement caused confusion and realize it is not an accurate statement. We meant that there is thought to be "atrialization" of the SV in reptiles and mammals Jensen et al, 2014 and for this to be some of the rationale for our lineage tracing. To be clear, our data are only for zebrafish and the lineage tracing data bears out that there is not significant contribution of atrial cardiomyocytes to the majority of the SV in WT embryos and non-CMs in the SV are not derived from embryonic atrial CMs in the mutant embryos. Our data does not inform us about the atrial segment only that CMs in the SV are not contributing to the adaptive response of the SV in the *nr2f1a* mutants. We have removed the statement about "atrialization" to alleviate any confusion and revised our rationale and presentation of the lineage tracing in the revised manuscript (lines 196-202).

6. Figure 2a, how come the bulbus cannot be seen on all images?

As the fish develop at these stages (7-28 days post-fertilization), the WT hearts are both looped and compact, which precluded imaging the whole hearts from the WT embryos when imaging the SV. Furthermore, the presence of the atrium in WT hearts prevented imaging the SV while it was still attached to the ventricle. Therefore, for the last two timepoints (21 and 28 dpf) in Fig. 2a the atrium and SV were isolated from the ventricle and BA prior to staining and imaging. In contrast to the WT hearts, the *nr2f1a* mutant hearts are more linear at all stages, which is why the BA and ventricle were able to be imaged in these hearts at the designated stages, as they did not preclude imaging of these chambers. We apologize that these details were not mentioned before. The experimental details for imaging have been included in the Methods of the revised paper (line 1152-1159). Importantly, these details and the lack of ventricle and BA do not change the analysis or interpretations from the images.

Figure 2c-d, the indicated widths appear exaggerated to the actual wall thickness, and please use absolute units for the length measurements, say μm .

The units for all the graphs with length measurements within the manuscript have been changed to μm . However, we respectfully disagree that the widths were exaggerated in our measurements. It is not clear to us from our methodology even how this exaggeration would occur. Importantly, the use of arbitrary units vs actual length measurements does not change the data or interpretations.

At present, this analysis is short of convincing, and so are the similar analyses of Figure 3. Figure 2j, it appears to show that the *Nr2f1a* mutants have myocardium in the atrium, in contrast to what is stated on the basis of Figure 1a – please explain.

It is not clear to us from their statement what Reviewer 3 thinks is not convincing. The *nr2f1a* mutant hearts do have some *Amhc* staining at the venous pole of their heart, which is exactly what was stated in the original manuscript and shown in Figure 1c,d and Supplementary Fig. 3. However, as shown in these figures, the amount of *Amhc*⁺ cardiomyocytes is much reduced compared to WT hearts, as there is not an atrial chamber, and all *Amhc*⁺ cardiomyocytes co-expresses *Vmhc* (Fig. 1e and Supplementary Fig. 3). However, when *Amhc* was used in other figures, we did not co-stain for *Vmhc*, as this should not be necessary to show in all the histological sections. The *Amhc* staining was used in some images, for instance Fig. 2g and Fig. 3e, to better visualize the border between the SV and the myocardium.

In addition, the images of Fig 2c-j 3b lack anatomical context, it is unclear whether the authors are comparing similar regions.

The sections used for the analysis were 3 sagittal sections at the midline. Every effort was made to compare similar regions within the SV of all fish. We have added statements to the Methods and a schematic to clarify where sections were made and analyzed (lines 1095-1096 and Supplementary Fig. 22).

Fig 2d, 3c, and 3f: In these figures similar comparisons are made regarding the thickness of the SV wall. Why did the experiment shown in 3f call for significantly more samples than 2d and 3c?

While the effect of exercise on the *nr2f1a* mutant fish is significant, it was more subtle than the drug treatments. It is possible this could be due to the experimental set up in that the fish were not continuously exercised (once per day) and we began it at 3 weeks, unlike the pharmacological treatments where the fish were treated continuously, whether the treatments were started at 48 hpf or 2 weeks post-fertilization. Therefore, the exercise experiments required more data points to ensure there was an appropriately powered and meaningful difference between the unexercised and exercised *nr2f1a* mutant fish. The additional data points though do not change the interpretation of the experiment.

Why the experiments regarding remodeling performed at different developmental stages?

For the exercise experiments, we chose to begin exercising the *nr2f1a* mutant fish before we observed the enlargement of the SV, which were the same stages that the EdU was applied to assess proliferation, and analyze the effect after we observed the enlargement of the SV in unexercised fish. We reasoned these would be sufficient time points to determine if the increased blood flow from exercise could exacerbate the defect. However, we feel the time points of initiating the exercise for this experiment are somewhat arbitrary, as in we could have initiated at 28 dpf or later, and analyzed at one week or longer, and we would have still expected to observe the same effect. Nevertheless, we chose 21 dpf because this was a stage that we did not observe ostensible enlargement of the SV yet and 35 dpf because the *nr2f1a* mutant hearts are significantly enlarged by then. Also, due to the physical nature of the swim tunnel itself, the size of the fish must be taken into account. Larvae younger than 21 dpf are too small to swim against even the slowest current and would have been in danger of being injured by the turbine. Regardless, we do not feel that the current experimental approach changes the interpretation from these experiments.

7. Estradiol, a steroid hormone, must be expected to affect transcription in numerous tissues and, concerning shear stress, its effects are not likely to be specific. So while the treatment reveals a difference in response between the WT and mutant, I don't see how these experiments inform us on the role of shear stress and blood flow more generally in causing the phenotype.

As we stated in the text, estradiol is used in animal models of hypertension (Gilbert, Mathis, and Ryan 2014; Zhou et al. 2020; Xue et al. 2007; Liu et al. 2014). It is not clear to us what Reviewer 3 means by "specific" nor do they explicitly state how we should approach this issue experimentally to better test this hypothesis. While we agree that estradiol signaling may not be a specific downstream effector of Nr2f1a or function only in cells of the SV, if that is what they are getting at, we disagree that the use of pharmacological inhibitors would need to be specific only to the SV to test the hypothesis that shear stress is contributing to the adaptive response of the SV. Moreover, it is not clear experimentally there would be a direct way to mechanically reduce the shear stress. Vasodilation would be one method, even if indirect, of reducing shear stress in vascular tissues. Vasodilators are used clinically to treat hypertension in humans, irrespective of how direct their function is. While we cannot distinguish directness of action for estradiol or any vasodilators, in the revised manuscript we have included experiments with additional vasodilators that function independent of estrogen receptor signaling and are used clinically. Just like with estradiol treatments, treatment with these drugs did not rescue the congenital atrial defects in the *nr2f1a* mutants, but mitigates the thickening that occurs in the SV. We interpret these results to mean that the thickening of the SV walls is a hypertensive response that shares conserved mechanisms as those found in hypertension with humans and other animal models, and due at least in part to the shear stress from the improper blood flow in these hearts. We have included the additional data in the revised manuscript and discuss these observations more thoroughly (lines 231-241 and 417-439; Figure 3b,c and Supplemental Figure 7a-d).

8. Your single cell analysis seems to suggest that the transcriptome of the bulbus is more affected than that of the non-myocardial sinus venosus. We must assume that Nr2f1a is not expressed in arterial vessels, but have you validated this and if so how do you explain the pronounced effect of loss of Nr2f1a on the identity of the bulbus? Is it not the bulbus that becomes more sinus venosus like than the other way around? Again, having extrapericardial vasculature as reference point appears to be important. For the time being, your experiments are not set up to differentiate between the possibilities that the bulbus and sinus are convergent on a shared vascular phenotype versus that one, or both tissues, are becoming more like each other.

a. As was shown in Supplementary Fig. 13 (Supplementary Fig. 10 of the original manuscript), *nr2f1a* expression is primarily restricted to the venous SMCs and ECs. In the revised manuscript, we have also provided an in situ of *nr2f1a* in an adult WT heart, showing its expression in the atrium, where it is highest, and absence from the BA and ventricle (Supplementary Fig. 18). We also now specifically highlight the lack of arterial *nr2f1a* expression for rationale in comparing SMCs from the WT and *nr2f1a* mutant BA (lines 356-361).

b. Please see our response **c** to Reviewer #2's, comment 2 as the comment here about similarity of SMCs within the WT and *nr2f1a* mutant BA is almost identical.

c. As indicated above in our response to Reviewer #3's, comment 4, we have performed gene-set enrichment analysis comparing the ECs clusters from our analysis to available single cell data from vascular cells of zebrafish embryos. This analysis suggests the venous ECs from WT SV normally have an intermediate vascular EC identity (Fig. 4e; lines 286-294). However, a subpopulation of venous ECs within the mutant *nr2f1a* are also have arterial EC identity (Supplementary Fig. 17; lines 343-355).

In your principal component analysis, please clarify how much of the variance is explained by the two components.

The variance for PCA is now indicated in the panel (Fig. 1g). Note it was moved to a main panel, as suggested by Reviewer #1.

L58: Can you articulate the reasoning for the mutant name 'acorn worm'? Given you are trying to understand early vertebrate evolution it is a bit too close to confusing, I would suggest, to name your mutant after hemichordates – are you implying the mutant heart resembles a heart-like structure in acorn worms?

The reasoning for the name "acorn worm" for this allele was because of the shape of the embryonic mutant heart when it was identified in screening. Specifically, we felt the morphology of the ventricle and smaller atrium in this *nr2f1a* mutant allele reminded us of the proboscis and gill region of hemichordates. Naming mutants based on physical characteristic from screens is standard, at least when a lot of forward genetic screening was being performed. There was no implication by the naming of the allele or a preconceived idea that this mutant allele would be used for studies of heart evolution when it was identified. While we state the name of the allele, we primarily refer to the mutant as *nr2f1a* or *nr2f1a^{aco}* throughout and do not think referring to it this way should be confusing.

L114: "was overtly comparable" – I don't think I understand what this means.

We apologize that this statement was confusing. We meant that it was observational and did not actually measure the sizes of the SVs at these earlier stages because there did not appear to be a reason to. The text in the revised manuscript has been changed to clarify this statement (line 206-208).

You make reference to, but do not provide, Fig 1m, Fig 5i.

Thank you for catching these. We did not catch these references to figures when editing the original manuscript. These references have been corrected or removed from the revised manuscript.

REVIEWERS' COMMENTS

Reviewer #2 (Remarks to the Author):

The authors have addressed my initial comments, provided a more nuanced interpretation of their data, and improved the manuscript.

It remains debatable whether it is possible to draw conclusions regarding evolution of the inflow tract based on a two species comparison in which the inflow tract either represents the ancestral state or is derived / adapted / specialized. Several scenarios for inflow tract / atrial evolution can be conceptualized in addition to those mentioned in the discussion section. The authors posit "the BA and SV in vertebrates stemmed from BS like structures in chordate ancestors...". Just for the sake of the argument, maybe the ancestral inflow tract gave rise to the AVC when the tube was elongated by adding posterior progenitors to form atrium and inflow tract. The inflow tract (SV) may have acquired cardiac muscular composition that was subsequently lost in particular species such as zebra fish.

These remarks are just to illustrate that interpretation of these data in the context of evolutionary implications is difficult at best. The authors are invited to more clearly acknowledge these potential pitfalls and uncertainties.

Reviewer #3 (Remarks to the Author):

The revision is an improvement of the manuscript.

The evolutionary implications of the study, which is one of the two selling points of the manuscript, still seem quite speculative; in part this is necessarily so, because only two species are compared. In the context of evolutionary studies, a two species comparison can often be insufficient (Physiol Zool 67, 1994).

The second selling point of the study, is the extent to which nr2f1a aco mutant fish can be used as a model system of cardiovascular disease. It is a clear improvement that morphometrics, such as the key readout of wall thickness, are no longer in arbitrary units. Nonetheless, I am still having a hard time understanding what you have measured; for example graph 3f, of which you write "(n=4 fish per group). Individual points on the graphs represent 20 measurements averaged from different regions of the SV in a single section (3 sections per fish)." If I understand this correctly, then each group should have 12 (avg) data points, but the graph shows innumerable data points. Also, one could argue that it is the averages of each of the four fish in each group that should be compared statistically (N=4 vs N=4) and certainly not every measurement. Possibly, the statistical tests have treated every measurement as one experiment, which they clearly are not; if so, please re-run your statistical tests. It is not inconceivable that significance can be affected, and at least an absence of significant difference in SV thickness would be consistent with the absence of significant difference in swimming performance of the fish. Related to the wall thickness phenotype, the nr2f1a aco mutant fish at 35dpf, with and without swimming, seemingly have a thinner sinus venosus wall than at 28dpf (with (3c) or without DMSO treatment (2c)). Is there an ontogenetic thinning of the sinus venosus wall?

Some main figures, I would suggest, could be stronger if they incorporated more of the data from the Supplementary data. Specifically, Figure 1g the data on the bulbus arteriosus is only from the wild type fish, whereas SI Fig. 9 reports data on the bulbus arteriosus from the nr2f1a aco mutant fish as well. Fig 3d-f, reports on nr2f1a aco mutant fish that have or haven't been in the swim tunnel, whereas the comparable data for the wild type fish, which is arguably quite relevant, is not included.

Response to Reviewer Comments – Manuscript NCOMMS-22-16391A-Z:

We appreciate that the reviewers felt the 1st revision of the manuscript was much improved. Reviewers 2 and 3 still had a few remaining issues that needed to be addressed. We have provided a point-by-point response to each of their comments. Our responses are indicated below in blue following the Reviewers' original comments. Changes made to the text in the 2nd revision of the manuscript in direct response to these comments are indicated in gray within the manuscript and the location is specified below.

Reviewers' comments:

Reviewer #2 (Remarks to the Author):

The authors have addressed my initial comments, provided a more nuanced interpretation of their data, and improved the manuscript.

It remains debatable whether it is possible to draw conclusions regarding evolution of the inflow tract based on a two species comparison in which the inflow tract either represents the ancestral state or is derived / adapted / specialized. Several scenarios for inflow tract / atrial evolution can be conceptualized in addition to those mentioned in the discussion section. The authors posit "the BA and SV in vertebrates stemmed from BS like structures in chordate ancestors...". Just for the sake of the argument, maybe the ancestral inflow tract gave rise to the AVC when the tube was elongated by adding posterior progenitors to form atrium and inflow tract. The inflow tract (SV) may have acquired cardiac muscular composition that was subsequently lost in particular species such as zebra fish.

These remarks are just to illustrate that interpretation of these data in the context of evolutionary implications is difficult at best. The authors are invited to more clearly acknowledge these potential pitfalls and uncertainties.

Thank you for the suggestion. We have included a statement at the end of the Discussion acknowledging there are limitations of comparing few species for making evolutionary hypotheses (lines 608-610).

Reviewer #3 (Remarks to the Author):

The revision is an improvement of the manuscript.

The evolutionary implications of the study, which is one of the two selling points of the manuscript, still seem quite speculative; in part this is necessarily so, because only two species are compared. In the context of evolutionary studies, a two species comparison can often be insufficient (Physiol Zool 67, 1994).

The second selling point of the study, is the extent to which nr2f1a aco mutant fish can be used as a model system of cardiovascular disease. It is a clear improvement that morphometrics, such as the key readout of wall thickness, are no longer in arbitrary units. Nonetheless, I am still having a hard time understanding what you have measured; for example graph 3f, of which you write "(n=4 fish per group). Individual points on the graphs represent 20 measurements averaged from different regions of the SV in a single section (3 sections per fish)." If I understand this correctly, then each group should have 12 (avg) data points, but the graph shows innumerable data points. Also, one could argue that it is the averages of each of the four fish in each group that should be compared statistically (N=4 vs N=4) and certainly not every measurement. Possibly, the statistical tests have treated every measurement as one experiment, which they clearly are not; if so, please re-run your statistical tests. It is not inconceivable that significance can be affected, and at least an absence of significant difference in SV thickness would be consistent

with the absence of significant difference in swimming performance of the fish. Related to the wall thickness phenotype, the *nr2f1a* aco mutant fish at 35dpf, with and without swimming, seemingly have a thinner sinus venosus wall than at 28dpf (with (3c) or without DMSO treatment (2c)). Is there an ontogenetic thinning of the sinus venosus wall?

Thank you for catching this. There indeed should have been 12 data points for the analysis of Fig. 3f. We have analyzed the data for Fig. 3f correctly, in the manner indicated in the figure legend, and added the proper graph to Fig. 3. Note that we analyzed the data both averaging the sections, as in the other figures of the manuscript and the figure legend, and with each individual data point from all the sections of each fish. The incorrect graph in the figure represented all the individual measurements from the section (n=240 for each unexercised and exercised *nr2f1a* mutant fish). This graph was unintentionally added to the figure and we did not catch it. Both ways of analyzing the data support there is a significant increase in the thickness of the sinus venosus following exercise in the *nr2f1a* mutants.

We have changed the control images in Fig. 3e to avoid any confusion. While for this experiment the walls of the sinus venosus in *nr2f1a* mutants were a little thinner than recorded in other experiments, we attribute this to some inherent variability in the thickness of the sinus venosus within *nr2f1a* mutants between cohorts. This does not change the interpretation of the experiment that increased flow exacerbates the thickening of the sinus venosus in the *nr2f1a* mutants. We presently have no evidence for ontogenetic thinning of the sinus venosus wall.

Some main figures, I would suggest, could be stronger if they incorporated more of the data from the Supplementary data.

Specifically, Figure 1g the data on the bulbus arteriosus is only from the wild type fish, whereas SI Fig. 9 reports data on the bulbus arteriosus from the *nr2f1a* aco mutant fish as well.

Unfortunately, we cannot do this comparison nor do we see how this change would work in the manuscript. The figures referenced are different data sets. Figure 1g is bulk RNA-seq, whereas Supplementary Figure 9 is single cell RNA-seq. The interpretations for Fig. 1 do not necessitate bulk RNA-seq for the bulbous arteriosus of *nr2f1a*^{aco} mutant fish.

Fig 3d-f, reports on *nr2f1a* aco mutant fish that have or haven't been in the swim tunnel, whereas the comparable data for the wild type fish, which is arguably quite relevant, is not included.

Thank you for the suggestion. We did not perform this experiment because we respectfully disagree that it is relevant. Our rationale is that one would not expect to observe an adaptive response and thickening of the sinus venosus in exercised WT fish compared to unexercised WT fish, at least of the time frame that we performed the experiment, as their hearts function normally and can properly accommodate increase stress. Even if there was a minor effect, this result would not negate that increased sheer stress exacerbates the effect observed in the *nr2f1a* mutant sinus venosus.